# Fear conditioning biases olfactory sensory neuron frequencies across generations

Clara W Liff[1,2], Yasmine R Ayman[1], Eliza CB Jaeger[1], Avery Cardeiro[1], Hudson S Lee[1], Alexis Kim[1], Angelica Vina-Abarracin[1], Dianne-Lee KD Ferguson[1,3], Bianca J Marlin[1,2,3,4]*

[1]Mortimer B. Zuckerman Mind Brain and Behavior Institute, Columbia University, New York, United States; [2]Department of Neuroscience, Columbia University, New York, United States; [3]Howard Hughes Medical Institute, Columbia University, New York, United States; [4]Department of Psychology, Columbia University, New York, United States

*For correspondence:
bjm2174@columbia.edu

Competing interest: The authors declare that no competing interests exist.

## eLife Assessment

This study provides **solid** evidence that odor fear conditioning biases olfactory sensory neuron receptor choice in mice and that this bias is detectable in the next generation. The authors use rigorous histological and behavioral analyses, including unsupervised behavioral quantification, to support the conclusion that odor-specific sensory representations can be shaped by experience and partially transmitted across generations. While the behavioral effects in offspring are modest and the mechanistic basis of inheritance remains unresolved, the study offers an **important** and carefully executed contribution to understanding experience-dependent sensory plasticity and its intergenerational consequences.

**Abstract** The main olfactory epithelium initiates the process of odor encoding. Recent studies have demonstrated intergenerationally inherited changes in the olfactory system in response to fear conditioning, resulting in increases in olfactory sensory neuron frequencies and altered responses to odors. We investigated changes in the cellular composition of the olfactory epithelium in response to an aversive stimulus. Here, we achieve volumetric cellular resolution to demonstrate that olfactory fear conditioning increases the number of odor-encoding neurons in mice that experience odor-shock conditioning (F0), *as well as their unconditioned offspring* (F1). We demonstrate that the increase in F0 is due, in part, to the biasing of the stem cell layer of the main olfactory epithelium. A detailed analysis of F1 behavior revealed subtle odor-specific differences between the offspring of unconditioned and conditioned parents, despite the absence of an active aversion to the conditioned odor. Thus, we reveal intergenerational regulation of olfactory epithelium composition in response to olfactory fear conditioning, providing insight into the heritability of acquired phenotypes.

## Introduction

Olfactory fear conditioning in mice results in the persistent avoidance of the conditioned odor. Moreover, the number of olfactory sensory neurons (OSNs) that respond to the conditioned odor increases in the main olfactory epithelium (*Jones et al., 2008*). Strikingly, this increase in the number of specific sensory neurons is observed not only in conditioned F0 males, but also in their unconditioned

offspring (F1), despite never having been exposed to the conditioned odor (*Dias and Ressler, 2014*; *Aoued et al., 2019*; *Aoued et al., 2020*). This phenomenon of intergenerational epigenetic inheritance describes the transfer of information from one generation to the next without alterations to the sequence of the genome.

Transgenerational epigenetic inheritance, the transfer of information beyond the F1 generation, is responsible for several examples of non-Mendelian transmission in plants, fission yeast, and worms (*Greer et al., 2011*; *Schmitz et al., 2011*; *Rechavi et al., 2014*; *Yu et al., 2018*; *Moore et al., 2021*). In these organisms, molecular genetics has provided a detailed mechanistic understanding of the transmission of epigenetic information from parent to multiple generations of offspring (*Rando and Verstrepen, 2007*; *Fitz-James and Cavalli, 2022*). While the existence of transgenerational epigenetic inheritance in mammals remains controversial, many studies have demonstrated the intergenerational inheritance of phenotypes in mice (*Carone et al., 2010*; *Dietz et al., 2011*; *Dias and Ressler, 2014*; *Gapp et al., 2014*; *Huypens et al., 2016*; *Chen et al., 2016*; *Chan et al., 2020*; *Cunningham et al., 2021*; *Toussaint et al., 2022*). As mice rely heavily on their sense of smell, olfactory conditioning in a parent may provide future generations with an adaptive advantage: enhanced sensitivity to aversive sensory features in the parent's environment. The intergenerational inheritance of olfactory properties provides a tractable model for understanding mammalian epigenetic inheritance because we can ask how signals responsible for specific changes in the nose are transmitted to the gamete, and then to the offspring.

The mechanisms responsible for the intergenerational increase in odor-responsive OSN number following aversive conditioning are more readily addressed in the olfactory epithelium of F0 mice than in F1 progeny. Elucidation of these local signaling events may then provide insight into the more distant transmission of information to the gametes. The mature olfactory sensory epithelium undergoes constant neurogenesis throughout the life of vertebrates. In mice, the lifespan of a mature OSN is estimated to be 30 days, and new sensory neurons are continually generated by the division of basal stem cells and differentiation into mature OSNs (*Liberia et al., 2019*). Continuous neurogenesis suggests that increases in specific OSN populations following olfactory fear conditioning could result from the increased birth or enhanced survival rate of a specific OSN subtype.

Olfactory perception is initiated by the recognition of odors by a large repertoire of receptors in the MOE. In mice, each mature OSN expresses only one of 1400 olfactory receptor genes (*Buck and Axel, 1991*). A cell's receptor choice is semi-stochastic and is mediated by a unique mechanism of transactivation that delivers the necessary transcription factors to only one allele of a single receptor gene (*Chess et al., 1994*; *Shykind et al., 2004*; *Lomvardas et al., 2006*). Neurons expressing a given receptor are distributed within a spatially restricted region of the MOE and project with precision to spatially invariant glomeruli in the olfactory bulb. Each odor can interact with multiple distinct receptors, resulting in the activation of a unique ensemble of glomeruli. The recognition of an odor requires the integration of information passed from glomeruli to mitral and tufted cells in the olfactory bulb, and then to convergent downstream olfactory areas (*Price and Powell, 1970*; *Price, 1985*; *Chen et al., 2014*; *Diodato et al., 2016*). If the receptor choice in a developing OSN can be biased by salient odor associations in the environment, this would afford a mechanism to alter OSN frequencies in the MOE, and potentially the subsequent downstream encoding of the odor.

In this study, we performed quantification of OSNs after olfactory fear conditioning, corroborating the results of an increase in the OSNs responsive to the conditioned odor. Moreover, enhanced numbers of these specific OSNs are also observed in the F1 offspring of conditioned fathers. Additionally, we demonstrate that this increase in the F0 generation is specific to OSN subtypes expressing receptors that respond to the conditioned odor and not a global increase in OSN number. We further demonstrate that a biased increase in specific OSNs after learning is likely to result from the enhanced proliferation of specific OSNs, suggesting that biased receptor choice underlies this phenomenon in the parent and is epigenetically inherited by their offspring. Behavioral analysis of conditioned F0 mice at multiple timepoints revealed avoidance at day 21 but not 42 or 63, despite the persistence of an increase in the number of OSNs responsive to the conditioned odor. Similarly, the F1 offspring of conditioned F0 males exhibited an increase in the number of conditioned odor-responsive OSNs but did not demonstrate an active avoidance of the conditioned odor. However, in-depth behavioral analysis revealed odor-specific effects in F1 cohorts in which offspring of paired F0 males exhibit alterations in behavior in the context of their father's conditioned odor. Ultimately, our findings suggest

that heritable increases in OSN populations, although distinct from active avoidance behavior, may influence more nuanced behaviors.

## Results

### Olfactory fear conditioning leads to an increase in conditioned odor-responsive cells in parents (F0)

In initial experiments, we asked whether we could observe changes in the abundance of receptors responsive to an odor after olfactory fear conditioning. The olfactory receptor M71 responds to the odor acetophenone, whereas the receptor MOR23 responds to the odor lyral (*Touhara et al., 1999*; *Bozza et al., 2002*). Transgenic mice with targeted mutations at the M71 or MOR23 locus to express green fluorescent protein (GFP) allowed for determination of M71 and MOR23 OSN abundance. Male and female *Olfr151*$^{IRES-tauGFP/IRES-tauGFP}$ (M71-GFP) or *Olfr16*$^{IRES-tauGFP/IRES-tauGFP}$ (MOR23-GFP) mice were subjected to an aversive olfactory conditioning paradigm in which a 10 s presentation of the odor acetophenone or lyral was paired with 0.75 mA foot shock, five times daily for 3 consecutive days (*Figure 1B and C*). The unpaired control group received the same number of odor presentations but experienced a 60-s delay between odor presentation and foot shock (*Figure 1C*), while the naive control group received no conditioning. The MOEs of conditioned mice were surgically extracted 21 days after the initiation of conditioning and subjected to iDISCO+ optical tissue clearing to visualize M71- and MOR23-expressing OSNs in intact olfactory epithelia (*Figure 1B, D and E*; *Renier et al., 2016*). Next, we imaged the anterior region of cleared epithelia using light sheet microscopy and counted the number of M71 or MOR23 OSNs in a fixed volume of tissue using automated spot detection software (*Figure 1G and I*; *Videos 1–6*).

Importantly, both M71 and MOR23 OSNs are expressed in the same anterior region of the MOE, enabling consistent imaging and counting protocols for both OSN populations. Male and female M71-GFP mice paired with acetophenone exhibited a 33% increase in the number of M71 OSNs 21 days after the initiation of conditioning when compared to unpaired controls (*Figure 1H*; One-way ANOVA. p<0.0001. Tukey's multiple comparisons. Naive vs. F0 paired p<0.0001. F0 unpaired vs. F0 paired p<0.0001. n=12,11,12; *Figure 1—figure supplement 1*). Male and female MOR23-GFP mice conditioned with lyral exhibited a 39% increase in MOR23 OSNs when compared to unpaired controls (*Figure 1J*; One-way ANOVA. p<0.0001. Tukey's multiple comparisons. Naive vs. F0 paired p<0.0001. F0 unpaired vs. F0 paired p<0.0001. n=7,9,9.; *Figure 1—figure supplement 1*). These results corroborate previous studies and demonstrate that odor-shock pairing leads to an increase in OSN populations that respond to the conditioning odor.

### Fear conditioning-induced increases in conditioned odor-responsive cells are heritable (F1)

We next asked whether the increase in the number of odor-responsive OSNs following conditioning is inherited by naive offspring of conditioned male mice. Ten days after the initiation of conditioning, we bred F0 males from both the unpaired and paired groups with naive female mice of matching genotypes (M71-GFP or MOR23-GFP). Each mating pair was separated ten days after co-housing to ensure that the offspring were never exposed to the conditioned father. Thus, any observed differences in the offspring could be attributed to the contents of the F0 male's germline. We collected and optically cleared MOEs from male and female 8- to 10-week-old F1 offspring and compared the abundance of M71 or MOR23 OSNs. These F1 mice were never exposed to acetophenone or lyral, nor had they undergone olfactory fear conditioning. Nonetheless, we observed a 36% increase in M71 OSNs in male and female F1 mice from fathers that underwent paired fear conditioning with acetophenone (*Figure 1H*; Tukey's multiple comparisons. F1 unpaired vs. F1 paired p<0.0001. n=12,14.; *Figure 1—figure supplements 1 and 2*). A similar relative increase of 27% was observed in MOR23 OSNs in male and female F1 mice from fathers that underwent paired fear conditioning with lyral (*Figure 1J*; Tukey's multiple comparisons. F1 unpaired vs. F1 paired p=0.0002. n=6,6.; *Figure 1—figure supplement 1*). These results demonstrate the intergenerational epigenetic inheritance of an olfactory phenotype, namely an increase in specific OSNs in naive F1 offspring following aversive conditioning in F0.

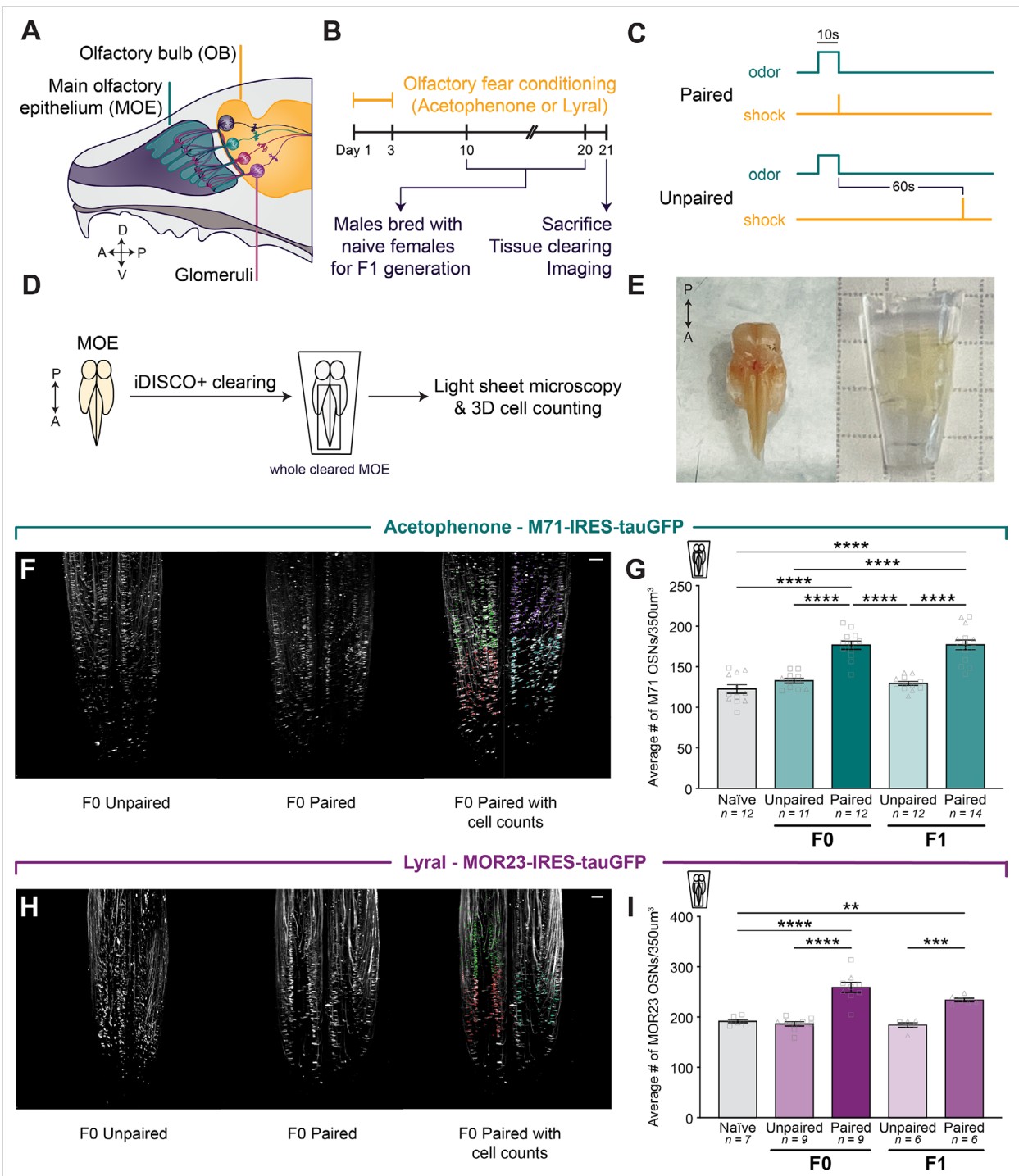

**Figure 1.** Olfactory fear conditioning leads to an increase in conditioned-odor-responsive cells in parents (F0) that is heritable (F1). (**A**) Schematic representation of the mouse main olfactory epithelium and olfactory bulb. MOE: main olfactory epithelium. OB: olfactory bulb. (**B**) Timeline of olfactory fear conditioning, breeding for the F1 generation, and MOE collection. (**C**) Experimental paradigms for olfactory fear conditioning groups. Mice in the paired condition received a foot shock that co-terminated with odor presentation, while mice in the unpaired condition received a foot shock 60 s after odor presentation. (**D**) Schematic demonstrating the process by which cells of interest in the MOE were quantified. Epithelia from both Olfr151[IRES-tauGFP/IRES-tauGFP] (M71-GFP) and *Olfr16*[IRES-tauGFP/IRES-tauGFP] (MOR23-GFP) adult mice were cleared using the iDISCO+ tissue-clearing protocol. Samples were imaged on a light sheet microscope and analyzed using Imaris spot detection software. (**E**) Images of the MOE before (left) and after (right) optical tissue clearing. (**F**) Example images of M71 OSNs in zone 1 of cleared MOE from both the unpaired (left) and paired (middle) conditions. Example image of an MOE with the counted cells represented by colored dots (right). Each set of colors represents a distinct counting cube. Scale bar: 200 μm. (**G**) The average number of M71 OSNs in a 350³ μm³ cube of epithelium of naive (gray), acetophenone unpaired (lighter green), and acetophenone paired

*Figure 1 continued on next page*

*Figure 1 continued*

(darker green) conditions in F0 and F1 (Error bars are standard error. One-way ANOVA. p<0.0001. Tukey's multiple comparisons. Naive vs. F0 paired p<0.0001. F0 unpaired vs. paired p<0.0001. Naive vs. F1 paired p<0.0001. F1 unpaired vs. paired p<0.0001. n=12,11,12,12,14.). Squares indicate males, triangles indicate females. (**H**) Example images of MOR23 OSNs in zone 1 of cleared MOE from both the unpaired (left) and paired (middle) conditions. Example image of an MOE with the counted cells represented by colored dots (right). Scale bar: 200 µm. (**I**) The average number (+/- standard error) of MOR23 OSNs in a $350^3$ µm$^3$ cube of epithelium in naive (gray), lyral unpaired (lighter purple), and lyral paired (darker purple) conditions in F0 and F1 (Error bars are standard error. One-way ANOVA. p<0.0001. Tukey's multiple comparisons. Naive vs. F0 paired p=<0.0001. F0 unpaired vs. paired p<0.0001. F1 unpaired vs. paired p=0.0368. n=7,9,9,6,6.).

The online version of this article includes the following figure supplement(s) for figure 1:

**Figure supplement 1.** Conditioned odor-responsive OSN counts by sex.

**Figure supplement 2.** F1 conditioned odor-responsive OSN counts by litter.

## Increased OSN abundance is specific to OSN subtypes responsive to the conditioned odor

In order to understand how a parental phenotype can be transmitted intergenerationally to future offspring, we decided to first focus on the mechanism underlying the parental phenotype, as it may serve as the basis for the intergenerational mechanism. To investigate the increase in cell number in the parental generation (F0), we first performed a series of control experiments to demonstrate that fear conditioning does not simply lead to a global increase in the number of OSNs in the MOE. First, we took advantage of the fact that the odor propanol does not activate M71 OSNs (*Jones et al., 2008*; *Johnson and Michael, 2000*). We repeated the same olfactory conditioning paradigm with homozygous M71-GFP mice, except with propanol as the conditioned odor, and counted the number of M71 OSNs in iDISCO-cleared intact MOEs (*Figure 2A*). We observed no difference in the number of M71 OSNs between the two groups of mice, indicating that an increase in OSN subtype depends on the conditioned odor activating that OSN population (*Figure 2D*; Student's unpaired t-test. Unpaired vs. paired p=0.3009. n=6,7.).

Next, we generated an *Olfr151*[IRES-tauRFP2] transgenic mouse line that expresses a tau-fused red fluorescent protein (RFP) in OSNs that express the M71 olfactory receptor (*Figure 2B*). To generate mice with an internal control, we crossed *Olfr151*[IRES-tauRFP2/IRES-tauRFP2] (M71-RFP) mice with MOR23-GFP mice to generate an *Olfr151*[IRES-tauRFP2/IRES-tauRFP2]; *Olfr16*[IRES-tauGFP/IRES-tauGFP] line in which all M71 OSNs are labeled red and all MOR23 OSNs are labeled green (*Figure 2C*). We conditioned these mice with acetophenone and counted the ratio of M71 OSNs to MOR23 OSNs in serial MOE sections. If olfactory fear conditioning leads to a global increase in the number of OSNs in the MOE, we would expect the ratio of M71 to MOR23 OSNs to remain unchanged following conditioning. However, we observed a significant increase in the ratio of M71 to MOR23 OSNs in paired mice only, demonstrating that odor-shock pairing increased the abundance of M71 OSNs relative to a population that does not respond to the conditioned odor (*Figure 2E*; One-way ANOVA. p=0.0062. Tukey's multiple comparisons. Naive vs. paired p=0.0097. Unpaired vs. paired p=0.0163.

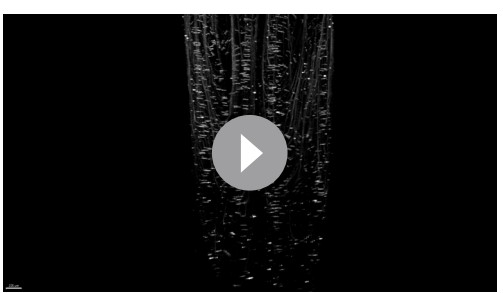

**Video 1.** Representative video of zone 1 of a cleared MOE from a homozygous M71-GFP acetophenone unpaired mouse. M71 OSNs are visualized in white. Scale indicated in video.

https://elifesciences.org/articles/92882/figures#video1

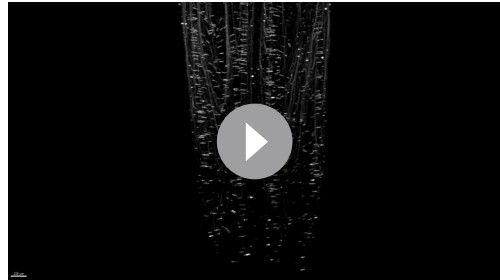

**Video 2.** Representative video zooming into a nasal turbinate (zone 1) of a cleared MOE from a homozygous M71-GFP acetophenone unpaired mouse. M71 OSNs are visualized in white. Scale indicated in video.

https://elifesciences.org/articles/92882/figures#video2

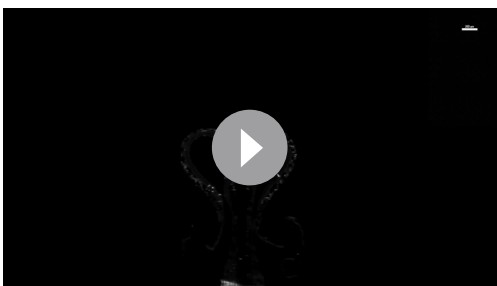

**Video 3.** Representative video of a cleared MOE from a homozygous M71-GFP acetophenone unpaired mouse, showing coronal slices of a section of zone 1 from anterior to posterior. The top of the video is dorsal, and the bottom is ventral. Scale indicated in video.

https://elifesciences.org/articles/92882/figures#video3

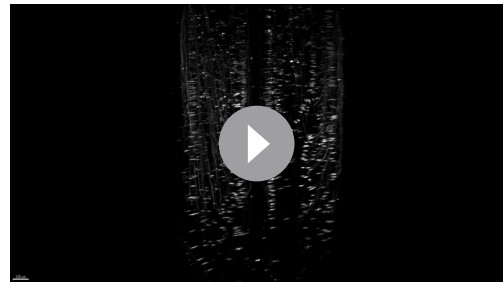

**Video 4.** Representative video of zone 1 of a cleared MOE from a homozygous M71-GFP acetophenone paired mouse. M71 OSNs are visualized in white. Scale indicated in video.

https://elifesciences.org/articles/92882/figures#video4

n=2,4,4.). Taken together, these experiments indicate that olfactory fear conditioning results in a *specific* increase in the number of cells responsive to the conditioned odor.

## Olfactory fear conditioning biases olfactory receptor choice toward conditioned-odor-responsive cell-specific identities

The olfactory epithelium exhibits continuous neurogenesis in mice. This constant turnover of OSNs suggests a possible mechanism for the increase in specific OSN populations responsive to the conditioned odor. The increase in M71 and MOR23 cells following odor-shock pairing could result from a biased increase in either the birth or survival rate of specific OSN subtypes. In initial experiments, we examined the relative number of M71 and MOR23 OSNs born during and immediately after olfactory fear conditioning. We injected homozygous M71-GFP and MOR23-GFP mice intraperitoneally with 5-Ethynyl-2'-deoxyuridine (EdU), a thymidine analog that incorporates into newly synthesized DNA and labels newborn cells, during each of the 3 days of training and for 2 subsequent days (*Figure 3A*). On day 21, we quantified the number of EdU-labeled M71 and MOR23 OSNs in MOE sections (*Figure 3A*). Since EdU has a half-life of approximately 35 min (*Cheraghali et al., 1995*), analysis of EdU 16 days after the cessation of injections reflects a pulse-chase, allowing us to quantify a subset of the neurons born during and 2 days following olfactory fear conditioning (*Figure 3B*).

The number of newborn M71 cells (EdU-labeled) out of total M71 cells was 1.24%±0.29 in naive mice, 2.91%±0.56 in acetophenone-unpaired mice, and 7.61%±0.53 in acetophenone-paired mice (*Figure 3D*; One-way ANOVA. p<0.0001. Tukey's multiple comparisons. Naive vs. paired p<0.0001. Unpaired vs. paired p<0.0001. n=6,6,6.). When lyral was used as the conditioned odor, the number of newborn (EdU-labeled) MOR23 cells out of total MOR23 cells was 0.29%±0.06 in naive mice, 0.55%±0.09 in lyral-unpaired mice, and

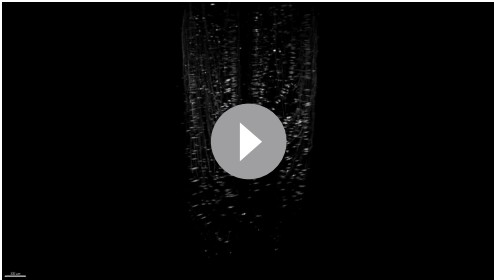

**Video 5.** Representative video zooming into a nasal turbinate (zone 1) of a cleared MOE from a homozygous M71-GFP acetophenone paired mouse. M71 OSNs are visualized in white. Scale indicated in video.

https://elifesciences.org/articles/92882/figures#video5

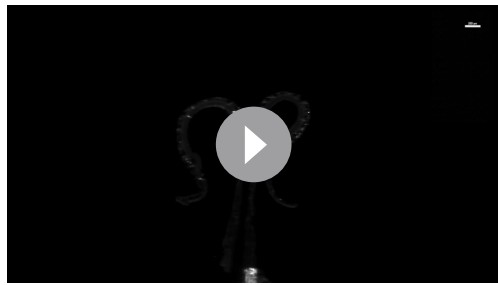

**Video 6.** Representative video of a cleared MOE from a homozygous M71-GFP acetophenone paired mouse, showing coronal slices of a section of zone 1 from anterior to posterior. The top of the video is dorsal, and the bottom is ventral. Scale indicated in video.

https://elifesciences.org/articles/92882/figures#video6

1.11%±0.21 in lyral-paired mice (*Figure 3E*; One-way ANOVA. p=0.0120. Tukey's multiple comparisons. Naive vs. paired p=0.0154. Unpaired vs. paired p=0.0653. n=4,6,8.). For both conditioning odors, we observed a significant increase in the number of newborn odor-responsive OSNs in mice that underwent paired conditioning (*Figure 3D and E*). These observations demonstrate that olfactory fear conditioning with acetophenone and lyral results in a significant increase in the number of newborn M71 and MOR23 cells, respectively. Despite subtype-specific differences in proliferation rates, odor-shock pairing significantly increased the number of newborn cells differentiating into an OSN subtype that responds to the conditioned odor.

## Conditioned-odor-responsive cell increase, but not avoidance behavior, is sustained through at least 9 weeks of cell turnover

We observe an increase in M71 OSNs 21 days after the start of acetophenone-paired olfactory fear conditioning. Given the continuous regeneration of the MOE, we next asked if this increase persists at later time points (*Figure 4A*). Since the half-life of the mouse MOE (the amount of time required for half of the epithelium to regenerate) is approximately 26 days (*Holl, 2018*), then at 42 days (6 weeks), approximately 67% will have been replaced by newly born neurons, and at 63 days (9 weeks), approximately 81% will have been replaced. At day 42, 3 weeks after the time point at which M71 counts were performed in initial experiments, we extracted the MOEs of male and female homozygous M71-GFP mice that underwent either unpaired or paired conditioning paradigms and optically cleared the tissue to compare M71 OSN abundance (*Figure 4C*). At the 42-day time point, we observed a 20% increase in the abundance of M71 OSNs in paired animals compared to controls (*Figure 4C*; Tukey's multiple comparisons. 42d unpaired vs. paired p=0.0476. n=8,8. *Figure 4—figure supplement 1*). At 63 days, we observed a 30% increase in M71 OSNs in the paired group compared to controls (*Figure 4C*; Tukey's multiple comparisons. 63d unpaired vs. paired p=0.0011. n=4,6.; *Figure 4—figure supplement 1*). The observation that these changes persist for at least 63 days after the start of conditioning, together with the reported 26 day half-life for the MOE, suggests the persistence of a signaling mechanism past the experience of olfactory fear conditioning itself.

To test whether a behavioral phenotype depends on OSN abundance, we performed an avoidance assay at the 42- and 63-day timepoints in a new cohort of mice. Mice were allowed 10 min to freely explore a three-chamber arena (trichamber) in which the conditioned odor (acetophenone) was continuously delivered through a port on one side, and a control odor (propanol) on the other (*Figure 4D*; *Video 7*). We calculated an approach-avoid index as the difference in the amount of time spent on the conditioned odor side compared to the control odor side, normalized by the total time spent exploring either chamber (*Figure 4D*). A positive index indicates overall approach towards the conditioned odor, whereas a negative index indicates an aversion to the conditioned odor. At the 42-day time point, we observed no significant difference in the avoidance indices of unpaired and paired mice, with both groups exhibiting a slight aversion to acetophenone (*Figure 4E*; Student's unpaired t-test. 42d unpaired vs. paired p=0.2248. n=10,11.). At the 63-day time point, we also observed no significant difference between the unpaired and paired mice (*Figure 4F*; Student's unpaired t-test. 63d unpaired vs paired p=0.8987. n=6,8.). Interestingly, when accounting for sex, we found a significant aversion in female paired mice at the 42-day timepoint that was not present at the 63-day timepoint and not present at either timepoint in males (*Figure 4—figure supplement 2*). Despite the behavioral sex difference at day 42, we did not observe a significant sex difference in M71 OSN counts at the same timepoint (*Figure 4—figure supplement 1*), indicating that the difference in behavior was not driven by a difference in MOE composition. Moreover, the absence of a strong behavioral aversion to acetophenone at 6- and 9 weeks post-conditioning despite a persistent increase in M71 representation in the MOE further suggests that an increase in OSN subtype following conditioning does not directly drive avoidance behavior. If this were the case, we would expect that the paired groups, which both exhibit heightened M71 counts at 42 and 63 days, would exhibit greater aversion to acetophenone than unpaired mice. Altogether, these results indicate that the mechanism underlying cellular adaptations in the MOE and the mechanism driving avoidance are independent.

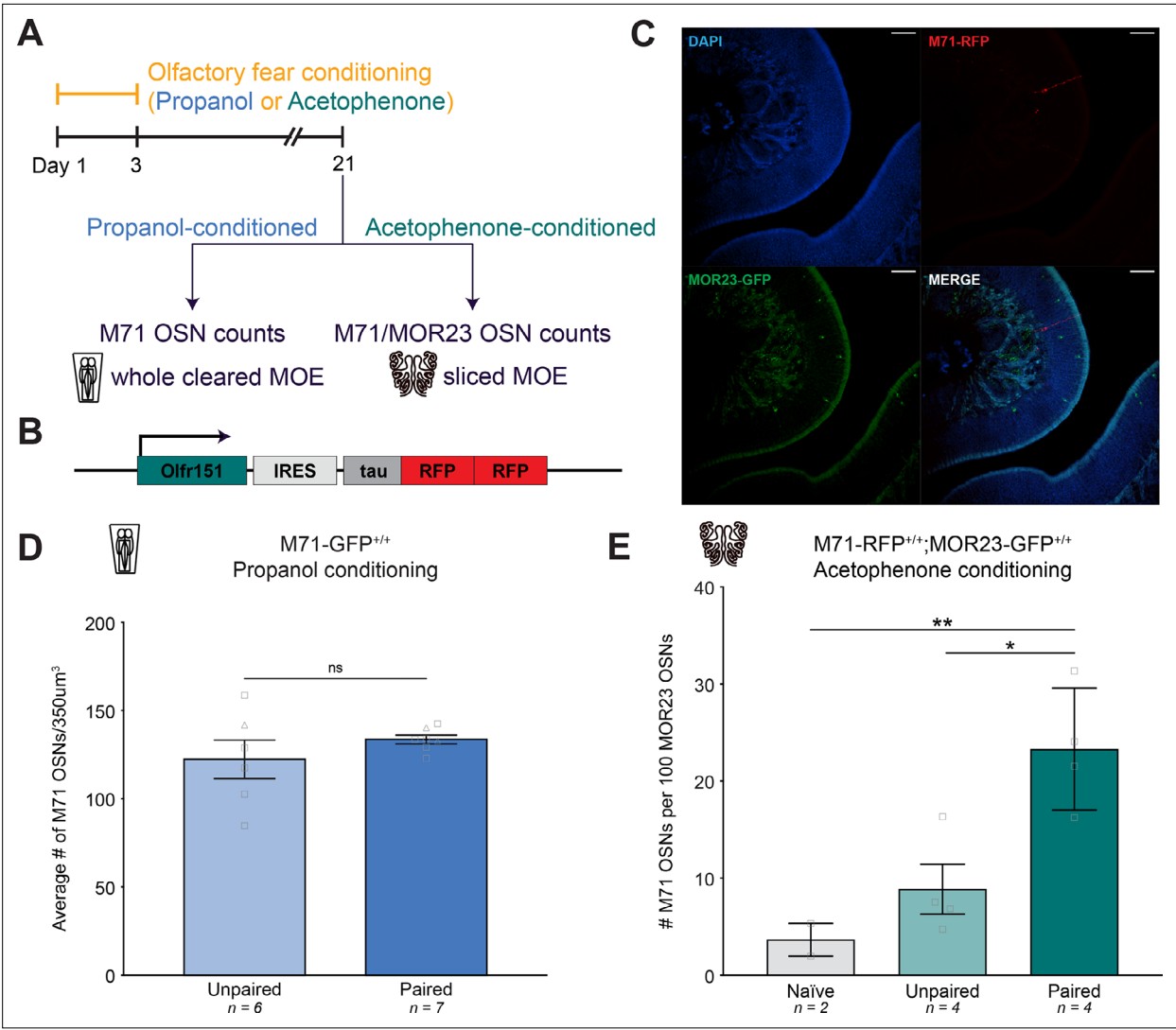

**Figure 2.** Increased OSN abundance is specific to OSN subtypes responsive to the conditioned odor. (**A**) Timeline of olfactory fear conditioning and MOE collection for both experiments. (**B**) Olfr151$^{IRES-tauRFP2/IRES-tauRFP2}$ (M71-RFP) targeted mutation. (**C**) Representative images showing DAPI (blue, top left), endogenous RFP in M71 OSNs (red, top right), endogenous GFP in MOR23 OSNs (green, bottom left), and the merged channels (bottom right) in a homozygous M71-RFP;MOR23-GFP animal. Scale bar: 50 μm. (**D**) The average number of M71 olfactory sensory neurons in a $350^3$ μm$^3$ cube of the epithelium in the propanol unpaired (light blue) and propanol paired (dark blue) conditions (Error bars are standard error. Student's unpaired t-test. Unpaired vs. paired p=0.3009. n=6,7.). Squares indicate males, triangles indicate females. (**E**) The ratio of M71 OSNs to 100 MOR23 OSNs in homozygous M71-RFP;MOR23-GFP mice following no conditioning (naive, gray), unpaired (light green), or paired (dark green) olfactory fear conditioning with acetophenone (Error bars are standard error. One-way ANOVA. p=0.0062. Tukey's multiple comparisons. Naive vs. paired p=0.0097. Unpaired vs. paired p=0.0163. n=2,4,4.).

## Olfactory fear conditioning leads to nuanced behavioral differences in F1 offspring

To further disentangle the relationship between increased OSN subtype representation and avoidance behavior, we performed extensive analyses on both F0 and F1 behavior. We generated three parallel F0 and F1 cohorts with three odors: acetophenone, lyral, and propanol. In each cohort of mice in which F0 males were conditioned for F1 breeding, we also conditioned male and female mice and tested them in the trichamber assay the day after conditioning (*Figure 5B*). We used ANY-maze tracking software to track the center position of the mice for calculation of approach-avoid indices, as well as to measure metrics like speed, distance traveled, and freezing. Importantly, males that were used for F1 breeding were not tested in the trichamber prior to breeding to avoid extinction, which has been shown to reverse the increased OSN phenotype in both F0 and F1 (*Aoued et al., 2019*).

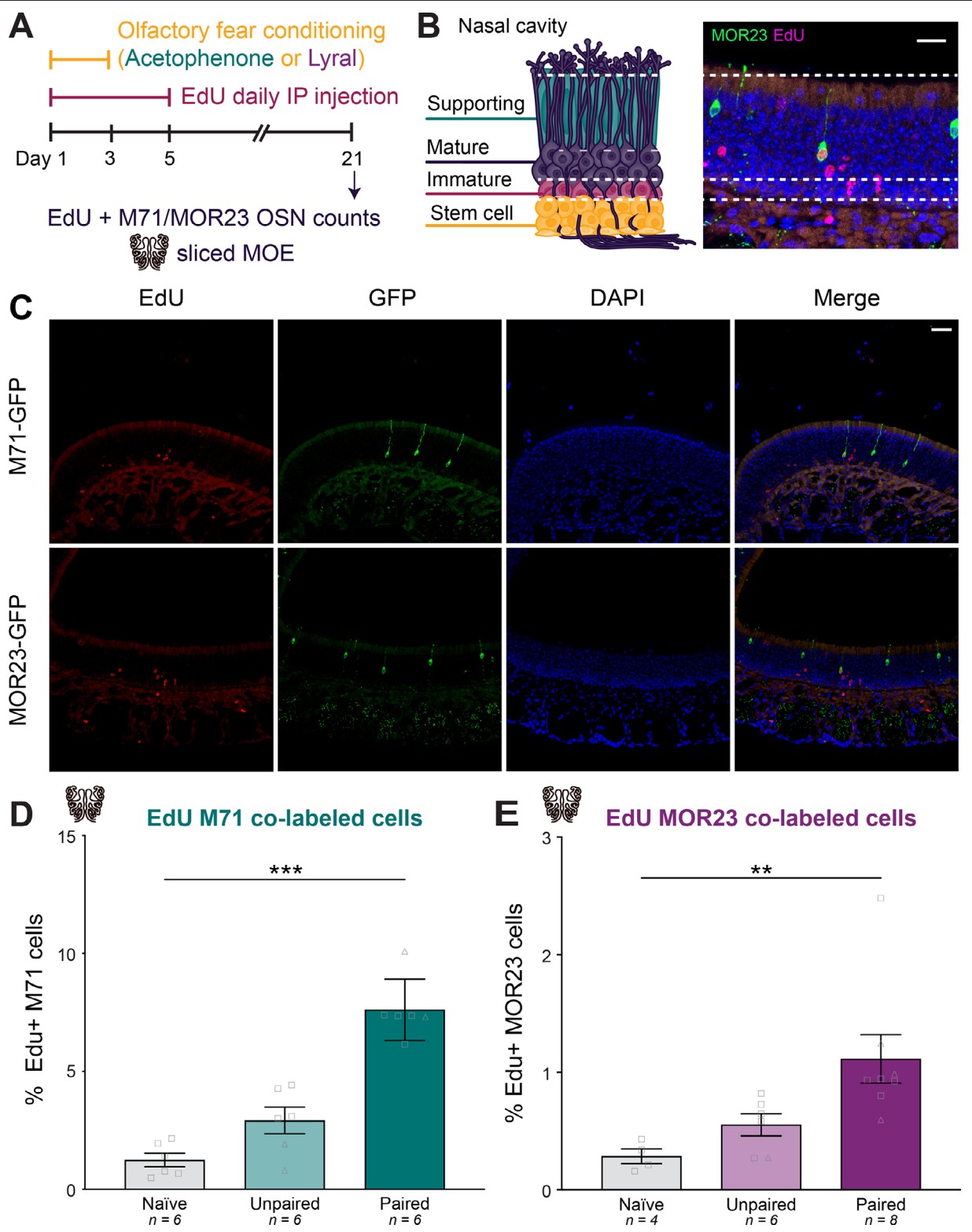

**Figure 3.** Olfactory fear conditioning biases olfactory receptor choice toward conditioned-odor-responsive cell-specific identities. (**A**) Timeline of olfactory fear conditioning, EdU injections, and MOE collection. (**B**) Schematic representation of the layers of the MOE, showing the stem cell, immature OSN, and mature OSN populations (left). Representative image of the MOE from a homozygous MOR23-GFP mouse showing EdU-positive cells (magenta) and a newborn (EdU+) MOR23 OSN (green and magenta). Scale bar: 20 μm. (**C**) Representative images showing MOE staining of EdU (red, first column), endogenous GFP (green, second column), DAPI (blue, third column), and the merged channels (fourth column) in homozygous M71-GFP

*Figure 3 continued on next page*

*Figure 3 continued*

and MOR23-GFP mice. Scale bar: 40 μm. (**D**) Percentage of EdU-positive M71 OSNs in naive, unpaired, and paired groups (Error bars are standard error. Kruskal-Wallis test. p<0.0001. Dunn's multiple comparisons. Naive vs. paired p=0.0009. Unpaired vs. paired p=0.0799. n=6,6,6.). Squares indicate males, triangles indicate females. (**E**) Percentage of EdU-positive MOR23 OSNs in naive, unpaired, and paired groups (Error bars are standard error. Kruskal-Wallis test. p=0.0003. Dunn's multiple comparisons. Naive vs. paired p=0.0040. Unpaired vs. paired p=0.0725. n=4,6,8.).

As expected, only mice in which odor and shock were paired exhibited robust aversion to the conditioned odor. F0 mice in the paired groups actively avoided the conditioned odor, whereas mice in the unpaired group exhibited no aversion to the conditioned odors (*Figure 5C, G and K*; Tukey's multiple comparisons. F0 acetophenone unpaired vs. paired p<0.0001. n=10,15. F0 lyral unpaired vs. paired p<0.0001. n=17,20. F0 propanol unpaired vs. paired p<0.0001. n=16,20.). Naive and unpaired mice spent roughly equal time exploring the control and conditioned odor chambers, whereas the paired mice spent an average of 67% (lyral-paired), 75% (acetophenone-paired), and 92% (propanol-paired) of the time exploring the control chamber.

Previous behavioral studies demonstrated that F1 offspring from fathers that experienced olfactory fear conditioning exhibit enhanced sensitivity to the conditioned odor in both odor potentiated startle and aversive odor association assays (*Dias and Ressler, 2014*). Therefore, we asked whether we could detect aversive behavioral responses in F1 populations after conditioning F0 fathers. We decided to use the trichamber assay as opposed to assays used in previous studies because we were interested in whether offspring inherited, in addition to an increase in conditioned odor OSN representation, a behavioral aversion to the odor. F1 offspring generated from conditioned males and naive females

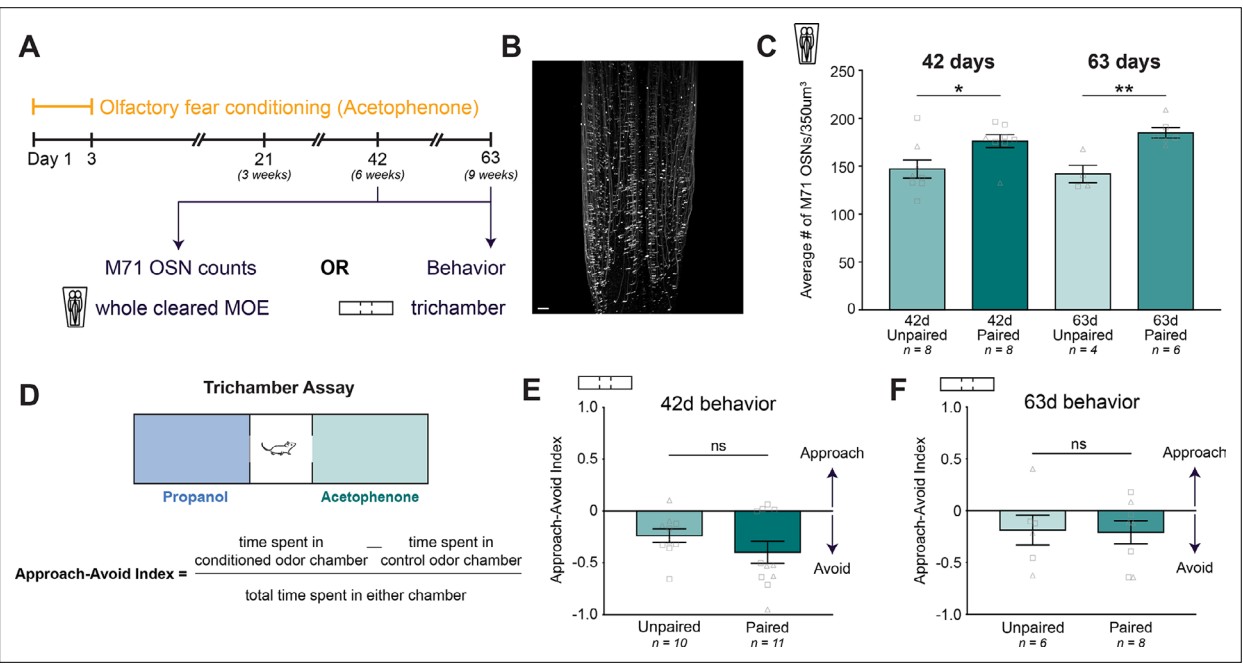

**Figure 4.** Conditioned-odor-responsive cell increase is sustained through at least 9 weeks of cell turnover. (**A**) Timeline of olfactory fear conditioning and extended MOE collection time points. (**B**) Example image of M71 OSNs in zone 1 of cleared MOE collected 42 days post-conditioning. Scale bar: 200 μm. (**C**) The average number of M71 OSNs in a $350^3$ μm$^3$ cube of epithelium of unpaired (light green) and paired (dark green) mice, 42- or 63 days post-conditioning (Error bars are standard error. One-way ANOVA. p=0.0033. Tukey's multiple comparisons. 42d unpaired vs. paired p=0.0476. 63d unpaired vs. paired p=0.0203. n=8,8,4,6.). Squares indicate males, triangles indicate females. (**D**) Schematic of the trichamber behavioral approach-avoidance assay and equation for the approach-avoid index. Positive values indicate approach to the conditioned odor (acetophenone), while negative values indicate avoidance. (**E**) The approach-avoid indices of unpaired and paired F0 mice at day 42 (Error bars are standard error. Student's unpaired t-test. 42d unpaired vs. paired p=0.2248. n=10,11.). Squares indicate males, triangles indicate females. (**F**) The approach-avoid indices of unpaired and paired F0 mice at day 63 (Error bars are standard error. Student's unpaired t-test. 63d unpaired vs paired p=0.8987. n=6,8.).

The online version of this article includes the following figure supplement(s) for figure 4:

**Figure supplement 1.** Extended timepoint OSN counts by sex.

**Figure supplement 2.** Extended timepoint trichamber assay by sex.

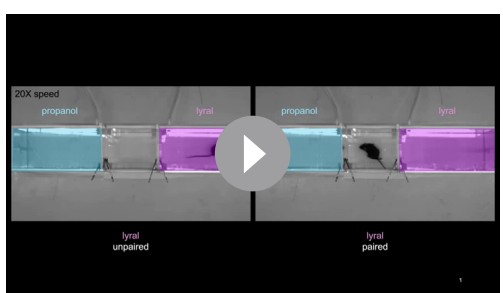

**Video 7.** Representative videos of odor preference behavior assay. Left chamber: propanol. Right chamber: lyral. In the left video, the mouse has undergone unpaired conditioning with lyral. In the right video, the mouse has undergone paired conditioning with lyral. Both videos are at 20X playback speed.

https://elifesciences.org/articles/92882/figures#video7

were assayed at 8- to 10 weeks of age in the same trichamber assay as the F0s. Importantly, none of the offspring had ever been exposed to the odor with which their father had been conditioned. We did not observe a significant difference in the avoidance indices of unpaired and paired F1 mice for any of the odors (*Figure 5C, G and K*; Tukey's multiple comparisons. Acetophenone F1 unpaired vs. paired p=0.9999. n=22,18. Lyral F1 unpaired vs. paired p=0.9976. n=6,13. Propanol F1 unpaired vs. paired p=0.7935. n=13,13. *Figure 5—figure supplements 1–3*).

Suspecting that our approach-avoid index may not be sensitive to more subtle behavioral differences, we performed additional behavioral analyses. Generation of spatial heat maps for the assays failed to reveal any clear qualitative differences in occupancy between unpaired and paired F1s in any of the three cohorts (*Figure 5D, H and L*). We also quantified freezing behavior during the trials and found no differences between unpaired and paired F1s in any of the cohorts (*Figure 5F, J and N*). However, we identified nuanced differences between unpaired and paired F1s that depended on the father's conditioned odor. When we quantified the distance traveled during the assay, we found that the F1s of lyral-paired fathers exhibited a hyperactive phenotype, traveling an average of 24.74±0.74 meters during the 10-min trial compared to 19.12±1.66 m in F1 lyral-unpaired mice (*Figure 5I*; S5E. Tukey's multiple comparisons. Lyral F1 unpaired vs. paired p=0.0325. n=6,13.). Conversely, the F1s of propanol-paired fathers exhibited a hypoactive phenotype, traveling an average of 15.89±1.11 m compared to 23.33±1.95 m in F1 propanol-unpaired mice (*Figure 5M*; Tukey's multiple comparisons. Propanol F1 unpaired vs. paired p=0.0045. n=13,13.; *Figure 5—figure supplement 1*). The phenotypes observed in the F1s of both lyral-paired and propanol-paired fathers are similarly reflected in the mean speeds of the mice during the trial, both of which were significantly different between the F1 unpaired and paired groups (*Figure 5—figure supplement 3*; Tukey's multiple comparisons. Lyral F1 unpaired vs. paired p=0.0315. n=6,13. Propanol F1 unpaired vs. paired p=0.0051. n=13,13.). We observed no significant differences in either distance traveled or mean speed in the F1s of acetophenone-conditioned fathers (*Figure 5E*; *Figure 5—figure supplement 3*). These analyses demonstrate that while the offspring of paired fathers do not actively avoid their father's conditioned odor, odor-shock pairing leads to subtle behavioral differences in offspring, and that these differences are specific to the conditioned odor used.

## Unsupervised machine learning analysis identifies behavioral differences in F1

To follow up on the differences observed between the F1s of unpaired and paired mice, we utilized Keypoint-MoSeq, a tool that uses unsupervised machine learning to identify behavioral modules, called 'syllables', in an unbiased manner (*Weinreb et al., 2024*). First, we trained a SLEAP model to track eight keypoints (nose, right ear, left ear, neck, center, right leg, left leg, tail-base) on each mouse across the entire trial (*Figure 6B*; *Pereira et al., 2022*). We trained a Keypoint-MoSeq model on the tracking data from 99 10-min trichamber videos (30 F0 and 69 F1 videos), for a total of $1.78 \times 10^6$ frames, to identify behavioral syllables with a median duration of approximately 30 frames (1 s; *Figure 6B and C*). This allowed us to ask whether the differences we observed between lyral F1s and propanol F1s at the timescale of 10 min could be tied to specific behaviors at the timescale of seconds, rather than a summary of 10 min. In our first analysis, we grouped the F1s of acetophenone-, lyral-, and propanol-unpaired fathers, and grouped the F1s of acetophenone-, lyral-, and propanol-paired fathers, to investigate whether there are differences between unpaired and paired offspring that generalize across the conditioned odors. Strikingly, we identified four syllables (3, 5, 11, and 22) with a significant difference in relative frequency (usage) between unpaired and paired F1s (*Figure 6D*;

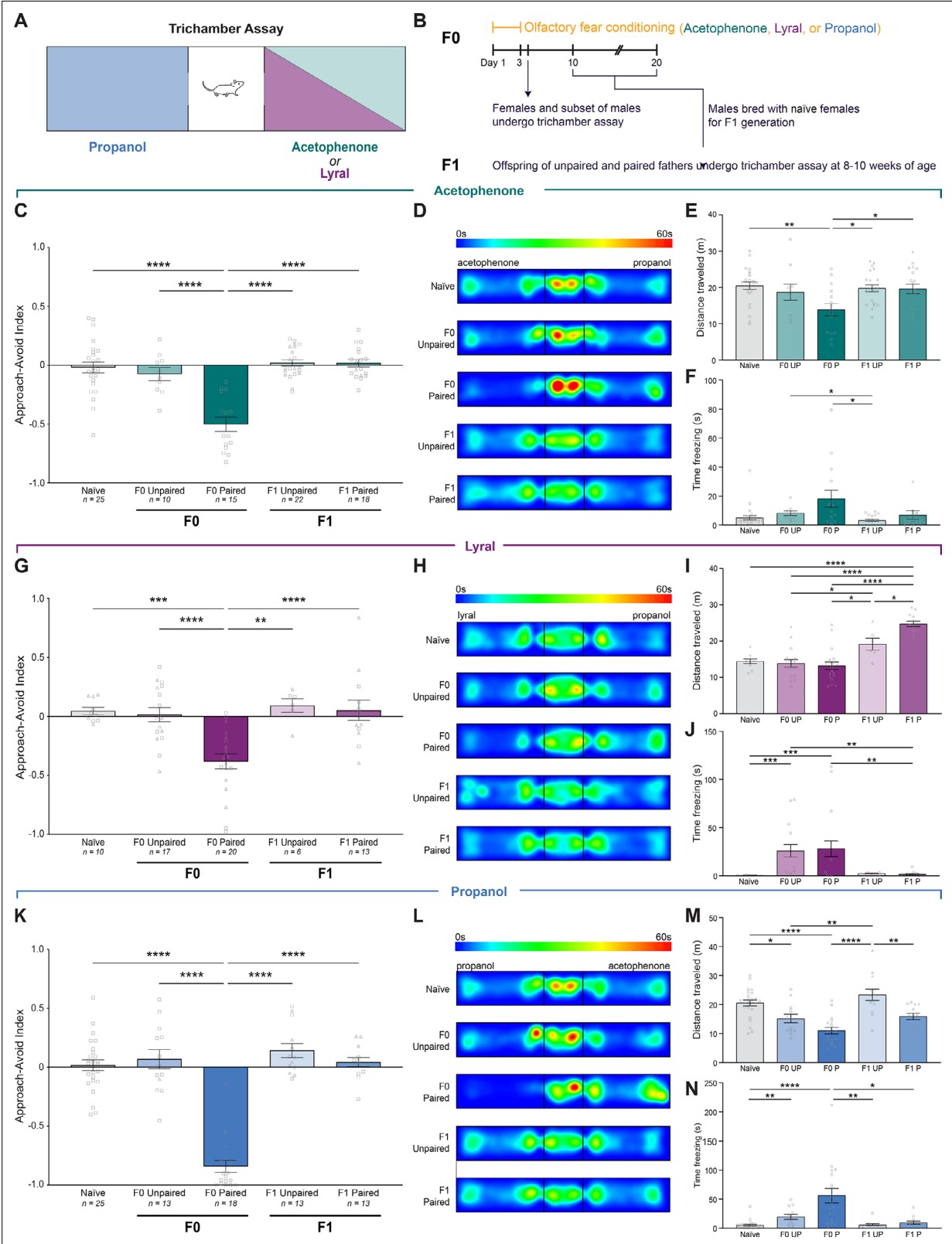

**Figure 5.** Olfactory fear conditioning leads to nuanced behavioral differences in F1 offspring. (**A**) Schematic of the trichamber assay showing the three conditioning odors and control odors. Propanol was the control odor for acetophenone and lyral, and acetophenone was the control odor for propanol. (**B**) Timeline of olfactory fear conditioning, behavior testing in F0, F0 breeding for the F1 generation, and F1 behavior testing. Conditioned F0 males used to breed for F1s did not undergo behavioral testing to prevent any extinction effects. (**C**) The approach-avoid indices of acetophenone-

*Figure 5 continued on next page*

*Figure 5 continued*

conditioned naive, unpaired, and paired F0 mice, and F1 mice bred from unpaired and paired F0 fathers (Error bars are standard error. One-way ANOVA. p<0.0001. Tukey's multiple comparisons. Naive vs. F0 Paired p<0.0001. F0 Unpaired vs. F0 Paired p<0.0001. F0 Paired vs. F1 Unpaired p<0.0001. F0 Paired vs. F1 Paired p<0.0001. n=25,10,15,22,18.). Squares indicate males, triangles indicate females. (**D**) Group-averaged heat maps for acetophenone-conditioned F0 mice and F1 offspring, with the acetophenone chamber on the left and the control (propanol) chamber on the right. (**E**) Distance traveled in the trichamber assay for acetophenone-conditioned F0 mice and F1 offspring (Error bars are standard error. One-way ANOVA. p=0.0041. Tukey's multiple comparisons. Naive vs. F0 Paired p=0.0032. F0 Paired vs. F1 Unpaired p=0.0141. F0 Paired vs. F1 Paired p=0.0103. n=25,10,15,22,18). (**F**) Time freezing for acetophenone-conditioned F0 mice and F1 offspring (Error bars are standard error. Kruskal-Wallis test. p=0.0022. Dunn's multiple comparisons. F0 Unpaired vs. F1 Unpaired p=0.0406. F0 Paired vs. F1 Unpaired p=0.0262. n=25,10,15,22,18.). (**G**) The approach-avoid indices of lyral-conditioned naive, unpaired, and paired F0 mice, and F1 mice bred from unpaired and paired F0 fathers (Error bars are standard error. One-way ANOVA. p<0.0001. Tukey's multiple comparisons. Naive vs. F0 Paired p=0.0004. F0 Unpaired vs. F0 Paired p<0.0001. F0 Paired vs. F1 Unpaired p=0.0013. F0 Paired vs. F1 Paired p<0.0001. n=10,17,20,6,13.). (**H**) Group-averaged heat maps for lyral-conditioned F0 mice and F1 offspring, with the lyral chamber on the left and the control (propanol) chamber on the right. (**I**) Distance traveled in the trichamber assay for lyral-conditioned F0 mice and F1 offspring (Error bars are standard error. One-way ANOVA. p<0.0001. Tukey's multiple comparisons. Naive vs. F1 Paired p<0.0001. F0 Unpaired vs. F1 Unpaired p=0.037. F0 Unpaired vs. F1 Paired p<0.0001. F0 Paired vs. F1 Unpaired p=0.0121. F0 Paired vs. F1 Paired p<0.0001. F1 Unpaired vs. F1 Paired p=0.0325. n=10,17,20,6,13.). (**J**) Time freezing for lyral-conditioned F0 mice and F1 offspring (Error bars are standard error. Kruskal-Wallis test. p<0.0001. Dunn's multiple comparisons. Naive vs. F0 Unpaired p=0.0001. Naive vs. F0 Paired p=0.0005. F0 Unpaired vs. F1 Paired p=0.0012. F0 Paired vs. F1 Paired 0.0041. n=10,17,20,6,13.). (**K**) The approach-avoid indices of propanol-conditioned naive, unpaired, and paired F0 mice, and F1 mice bred from unpaired and paired F0 fathers (Error bars are standard error. One-way ANOVA. p<0.0001. Tukey's multiple comparisons. Naive vs. F0 Paired p<0.0001. F0 Unpaired vs. F0 Paired p<0.0001. F0 Paired vs. F1 Unpaired p<0.0001. F0 Paired vs. F1 Paired p<0.0001. n=25,13,18,13,13.). (**L**) Group-averaged heat maps for propanol-conditioned F0 mice and F1 offspring, with the propanol chamber on the left and the control (acetophenone) chamber on the right. (**M**) Distance traveled in the trichamber assay for propanol-conditioned F0 mice and F1 offspring (Error bars are standard error. One-way ANOVA. p<0.0001. Tukey's multiple comparisons. Naive vs. F0 Unpaired p=0.0304. Naive vs. F0 Paired p<0.0001. F0 Unpaired vs. F1 Unpaired p=0.0014. F0 Paired vs. F1 Unpaired p<0.0001. F1 Unpaired vs. F1 Paired p=0.0045. n=25,13,18,13,13.). (**N**) Time freezing for propanol-conditioned F0 mice and F1 offspring (Error bars are standard error. Kruskal-Wallis test. p<0.0001. Dunn's multiple comparisons. Naive vs. F0 Unpaired p=0.0074. Naive vs. F0 Paired p<0.0001. F0 Paired vs. F1 Unpaired p=0.0011. F0 Paired vs. F1 Paired p=0.028. n=25,13,18,13,13.).

The online version of this article includes the following figure supplement(s) for figure 5:

**Figure supplement 1.** Trichamber assay metrics by sex.

**Figure supplement 2.** Trichamber assay avoidance index by litter.

**Figure supplement 3.** Additional trichamber assay metrics.

Kruskal-Wallis test with Dunn's multiple comparisons. F1 unpaired vs. paired. Syllable 3 p=0.0070. Syllable 5 p=0.0054. Syllable 11 p=0.0251. Syllable 22 p=0.0184.). Of those four syllables, syllables 3 and 5 were used significantly more in the F1 paired group compared to the F1 unpaired group, while syllables 11 and 22 were used significantly less in the F1 paired group (*Figure 6D*). We also noticed interesting qualitative differences in the spatial usage patterns of each syllable, with unpaired and paired F1s showing distinct usage distributions relative to their distance from the conditioned odor port (*Figure 6—figure supplement 1*). Next, we were interested in behavioral differences specifically in response to the conditioned odor of the parent. To this end, we narrowed the analysis to syllable usage only when the mouse was in the conditioned odor chamber. While both F1 unpaired and F1 paired mice spent similar amounts of time in the conditioned odor chamber (*Figure 6E*; *Figure 5— figure supplement 3*), the F1 paired group exhibited greater usage of syllables 5 and 17 compared to controls (*Figure 6F*; Kruskal-Wallis test with Dunn's multiple comparisons. F1 unpaired vs. paired. Syllable 5 p=0.0406. Syllable 17 p=0.0467.). Taken together, these findings point to small but significant behavioral differences between the offspring of unpaired and paired fathers that generalize across all three conditioning odors.

Lastly, we performed the same analysis separately for the three F1 cohorts of acetophenone-, lyral-, and propanol-conditioned F0 fathers. Interestingly, in parallel with the earlier analyses, we again observed no differences between the unpaired and paired F1s of acetophenone-conditioned fathers (*Figure 6G*) despite differences in M71 OSN representation. Also consistent with our earlier analyses, we found significant differences between the unpaired and paired F1s of lyral- and propanol-conditioned fathers, and the differences between the unpaired and paired F1 groups were unique to the odor (*Figure 6H and I*). We identified six syllables with significantly different frequencies between unpaired and paired F1s of lyral-conditioned fathers: 2, 3, 5, 7, 15, and 20 (*Figure 6H*; Kruskal-Wallis test with Dunn's multiple comparisons. Lyral F1 unpaired vs. paired. Syllable 2 p=0.0016. Syllable 3 p=0.0272. Syllable 5 p=0.0030. Syllable 7 p=0.0025. Syllable 15 p=0.0208. Syllable 20 p=0.0002.).

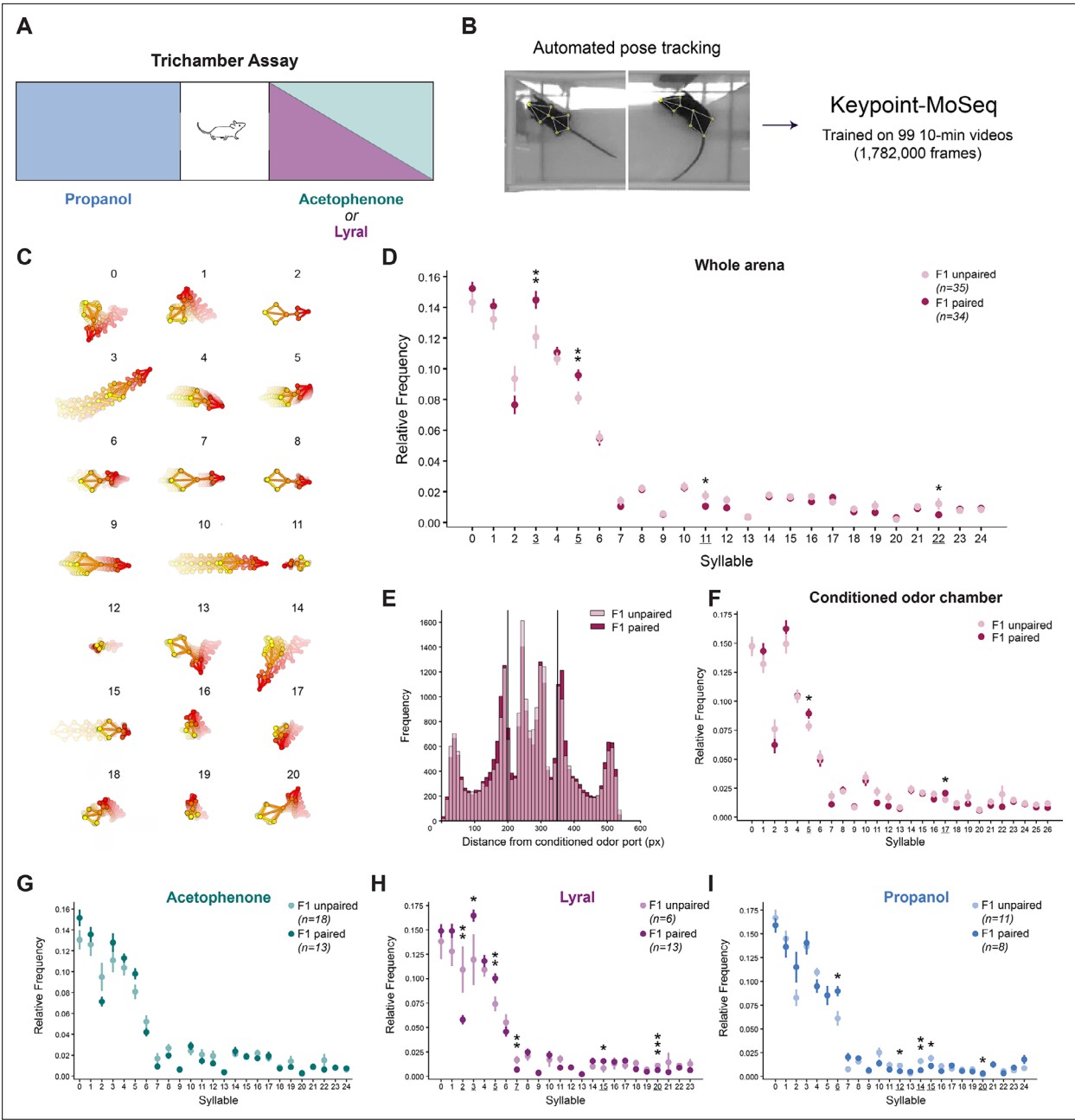

**Figure 6.** Unsupervised machine learning analysis identifies behavioral differences in F1. (**A**) Schematic of the trichamber assay showing the three conditioning odors and control odors. Propanol was the control odor for acetophenone and lyral, and acetophenone was the control odor for propanol. (**B**) Analysis pipeline for Keypoint-MoSeq. Eight key points were tracked across the entire 10-min trichamber assay for 99 videos, and the tracking data was used to train a Keypoint-MoSeq model. (**C**) Trajectory plots of the 21 most frequently used syllables across the dataset. (**D**) The relative usage frequencies of syllables in the whole trichamber arena in the F1 offspring of unpaired (light pink) and paired (dark pink) fathers. Syllables with significantly different usage between groups are underlined in the x-axis and denoted with asterisks above the data points (Error bars are standard error. Kruskal-Wallis test with Dunn's multiple comparisons. Syllable 3 F1 unpaired vs. F1 paired p=0.00696. Syllable 5 p=0.0054. Syllable 11 p=0.02505. Syllable 22 p=0.0184. n=35,34.). (**E**) Histogram of the frequency (number of observations divided by the bin width) of each animal's center point along the x-axis (x distance from the conditioned odor port). The two vertical bars indicate the divisions between the three chambers (conditioned odor chamber left, control odor chamber right). (**F**) The relative usage frequencies of syllables in the conditioned odor chamber in F1 unpaired (light pink) and F1 paired (dark pink) (Error bars are standard error. Kruskal-Wallis test with Dunn's multiple comparisons. Syllable 5 F1 unpaired vs. F1 paired p=0.0406. Syllable 17 *P*=0.046695. n=35,34.). (**G**) The relative usage frequencies of syllables in the whole arena in the F1 offspring of acetophenone-unpaired (light green) and acetophenone-paired (dark green) fathers (Error bars are standard error. n=18.13.). (**H**) The relative usage frequencies of syllables in the whole arena in the F1 offspring of lyral-unpaired (light purple) and lyral-paired (dark purple) fathers (Error bars are standard error. Kruskal-Wallis test with Dunn's multiple comparisons. Syllable 2 lyral F1 unpaired vs. F1 paired p=0.0016. Syllable 3 p=0.02724. Syllable 5 p=0.003. Syllable 7 p=0.00252.

*Figure 6 continued on next page*

*Figure 6 continued*

Syllable 15 p=0.0208. Syllable 20 p=0.00015. n=6,13.). (**I**) The relative usage frequencies of syllables in the whole arena in the F1 offspring of propanol-unpaired (light blue) and propanol-paired (dark blue) fathers (Error bars are standard error. Kruskal-Wallis test with Dunn's multiple comparisons. Syllable 6 propanol F1 unpaired vs. F1 paired p=0.0206. Syllable 12 p=0.04068. Syllable 14 p=0.0072. Syllable 15 p=0.03204. Syllable 20 p=0.0148. n=11,8.).

The online version of this article includes the following figure supplement(s) for figure 6:

**Figure supplement 1.** Density of syllable usage across space in F1 offspring.

The upregulated syllables in lyral paired F1s were higher-velocity syllables, while the downregulated syllables were relatively stationary/low-velocity, consistent with the hyperactive phenotype earlier described by greater distance traveled and higher mean speed in lyral-paired F1s. Between the unpaired and paired F1s of propanol-conditioned fathers, we found five syllables with significantly different frequencies: 6, 12, 14, 15, and 20 (*Figure 6I*; Kruskal-Wallis test with Dunn's multiple comparisons. Propanol F1 unpaired vs. paired. Syllable 6 p=0.0206. Syllable 12 p=0.0407. Syllable 14 p=0.0072. Syllable 15 p=0.0320. Syllable 20 p=0.0148.). Compared to the lyral F1s, the opposite was true for the differentially used syllables in the propanol F1s, in which the upregulated syllables in propanol-paired F1s were lower-velocity syllables, and the downregulated syllables were relatively high-velocity syllables, consistent with the hypoactive phenotype earlier described. Here, we have demonstrated that olfactory fear conditioning in the F0 population leads to conditioned odor-specific effects in the F1 offspring of paired mice.

## Discussion

We used tissue-clearing and light-sheet microscopy to demonstrate a specific increase in the number of OSNs expressing the receptor for an aversively conditioned odor. Enhanced abundance of OSNs responsive to the conditioned odor was observed in naive F1 offspring of conditioned F0 males. These results are consistent with studies that employ other cellular visualization techniques (*Dias and Ressler, 2014*; *Aoued et al., 2019*; *Aoued et al., 2020*). In F0, this increase is stable for at least 63 days, a time by which most of the cells present during olfactory fear conditioning have been replaced by newborn sensory neurons. The increase in OSN count in F0 results, in part, from the contribution of newborn neurons responsive to the conditioned odor, demonstrating a biasing of OSN development. The sustained increase in F0, along with the inheritance in F1, suggests that there is a stable signal that is responsible for the induction, maintenance, and inheritance of the increase in OSNs responsive to the paired odor.

The stochastic choice of olfactory receptors may provide an opportunity to alter the representation of receptors to allow an organism to adapt to the environment. Changes in the number of OSNs may lead to an increase in sensitivity of the paired odor. A change in OSN number may also lead to increased inputs to downstream sensory areas. Such perceptual changes have been reported in the motor, visual, olfactory, and auditory systems, where topographical arrangements at the primary sensory cortex are modulated in certain fear conditioning paradigms in mammals (*Ressler et al., 1994*; *Vassar et al., 1994*; *Mombaerts et al., 1996*; *Lai et al., 2018*; *Xu et al., 2019*; *Li et al., 2019*), although this has not yet been demonstrated intergenerationally. We observed an increase in conditioned odor-responsive neurons in both the F0 and F1 populations. Only the F0, however, exhibited active avoidance behavior. Interestingly, the F0 paired groups exhibited varying degrees of avoidance based on the odor used in conditioning (*Figure 5*). Similarly, the effects that we observed in the F1 offspring of paired mice depended on the conditioning odor of the father. This presents the possibility that factors such as odor volatility, the abundance of odor-responsive OSN populations, and the binding affinities of those populations may influence the behavioral phenotypes in both parent and offspring despite the shared cellular phenotype of increased OSN abundance. We speculate that the increase in neurons responsive to the conditioned odor could enhance the sensitivity to, or the discrimination of, the paired odor in F0 and F1. This would enable the F1 population to learn that odor predicts shock with fewer training cycles or less odor when trained with the conditioned odor.

These findings set a foundation to uncover the mechanism by which olfactory receptor bias is communicated within the main olfactory epithelium, to the germline, and, moreover, maintained during the development of offspring. What remains to be uncovered are the mechanisms to bias the choice of specific receptors in the main olfactory epithelium and how the information governing

the biasing of receptor choice is transferred to the gametes. In mice, the paternal transmission of epigenetic information has been observed following metabolic disturbances, social stress, and exposure to drugs and toxins (*Huypens et al., 2016*). High-fat or low-protein diets, as well as caloric restriction in the father, result in metabolic disturbances in the offspring, even after in vitro fertilization (*Carone et al., 2010*; *Chen et al., 2016*). Parental stressors, such as chronic defeat or maternal separation, result in hormonal disturbances and behavioral phenotypes in the offspring (*Dietz et al., 2011*; *Morgan and Bale, 2011*; *Gapp et al., 2014*). Finally, toxins and addictive drugs result in an array of metabolic disturbances in the F1 population that recapitulate the paternal state (*Toussaint et al., 2022*). These paternal stressors are associated with metabolic and hormonal disturbances that can readily act at a distance to affect the gamete. It has been demonstrated in male gametogenesis that extracellular vesicles in the testes transmit an RNA payload as they fuse with maturing sperm (*Rando, 2016*; *Sharma et al., 2018*; *Morgan et al., 2019*; *Chan et al., 2020*). Such studies provide insights into a mechanism by which an olfactory sensory experience paired with fear learning could transmit receptor-specific information from one generation to the next.

Controversies surrounding heritable behaviors in mammals hinge on the question of what biological adaptations can be inherited that would lead to alterations in the behavior of future generations. We focused on the ethological behavior of olfactory avoidance and observed nuanced behaviors in the F1 population that have not formerly been described. We hypothesize that these changes in behavior are related to the inheritance of an increase in specific OSN populations, but the question remains whether there are heritable changes in downstream brain circuits that could contribute to a behavioral phenotype. Multiple results lead us to believe that changes in OSN representation in the MOE are not sufficient to drive olfactory avoidance of the conditioned odor. First, we observe robust avoidance of the conditioned odor one day after olfactory fear conditioning, at which point we expect minimal effects of biased neurogenesis given the rates of OSN maturation and turnover in the MOE. Thus, the behavioral effects in F0 likely do not require an increase in conditioned odor-responsive OSN abundance. Second, we observe the persistence of an increase in M71 OSN representation at the 42- and 63-day timepoints, when mice no longer avoid the conditioned odor, indicating that higher cell number alone does not drive avoidance. Lastly, the F1 offspring of paired males do not actively avoid the conditioned odor despite exhibiting higher numbers of conditioned odor-responsive OSNs. This disentanglement of cell number and avoidance behavior leads to the hypothesis that, in parallel with the inheritance of the MOE phenotype, changes to brain circuits responding to stress may also be inherited. One possibility is that heritable changes in neuromodulator systems could shift the balance of internal states in offspring, as has been observed in rodent studies (*Champagne, 2008*; *Rodgers et al., 2013*; *Gapp et al., 2021*).

Our study elaborates on a function of sensory systems in which a learned adaptation can influence future generations. Thus, the distinction between innate and learned behaviors may be fundamentally flexible — learned adaptations in the parent may have the potential to become innate in their offspring. Understanding the mechanisms of inherited adaptation will provide insight for interventions when these changes no longer serve as adaptive to the organism.

## Materials and methods

### Mice

All procedures were approved by the Columbia University Institutional Animal Care and Use Committee under protocol #AABL8552. All mice were housed with a 12 hr light/12 hr dark cycle and fed ad libitum. *Olfr151*^IRES-tauGFP/IRES-tauGFP (Stock #006676), *Olfr16*^IRES-tauGFP/IRES-tauGFP (Stock #006643), and C57BL/6 J (Stock #000664) mice were obtained from The Jackson Laboratory or gifted from the Lomvardas and the Axel laboratories. The *Olfr151*^IRES-tauRFP2 line was generated in-house.

### Transgenic mouse line

The *Olfr151*^IRES-tauRFP2 mouse line was generated by homologous recombination of mouse ES cells using established techniques. Briefly, the targeting vector (Addgene, 15510) was linearized using PmeI and electroporated into MM13 ES cells (129 S/SvEv). Targeted ES clones were identified by Southern blot hybridization and positive ES cells were transferred into C57BL/6 blastocysts. Male chimeras were bred with C57BL/6 females to establish germline transmission and subsequently outcrossed to

C57BL/6 females for at least five generations. Deletion of the self-excising Neo cassette was confirmed by PCR of genomic DNA.

## Olfactory fear conditioning

8–12 week-old male and female mice were trained to associate acetophenone (Sigma-Aldrich, 42163), lyral (IFF, 00129214), or propanol (Sigma-Aldrich, I9516) with 0.75 mA foot shocks. Odors were diluted to 10% v/v in mineral oil (Fisher, O121-1). The mice were trained on 3 consecutive days, with each training day consisting of five presentations of odor for 10 s. For mice in the paired condition, the odor presentations were co-terminated with a 0.75 mA foot shock. For mice in the unpaired condition, there was a 60-s delay between the odor presentation and foot shock (*Figure 1C*). Olfactory fear conditioning boxes, olfactometers, and software were obtained from Med Associates. Mice were randomly assigned to experimental conditions.

## Tissue clearing

MOEs were perfused, dissected, and processed according to the iDISCO+ protocol (*Renier et al., 2016*). Whole MOEs were processed in 5 mL volumes. Samples were postfixed in 4% paraformaldehyde in 1X PBS (Electron Microscopy Sciences, 15,710 S) overnight at 4°C. The following day, they were washed with 1X PBS (3 × 30 min), gradually dehydrated with methanol (MeOH; Sigma-Aldrich, 322415) over 5 hr, and incubated in 66% dichloromethane (DCM; Sigma-Aldrich, 270997)/33% MeOH overnight. The samples were washed in 100% MeOH the following day, chilled at 4°C, bleached in 5% hydrogen peroxide (Sigma-Aldrich, 216763) in MeOH overnight at 4°C, and then gradually rehydrated the next day. Samples were permeabilized for 2 days and blocked in 6% goat serum (Jackson ImmunoResearch, 005-000-121) for 2 days at 37°C. Next, they were labeled with a 1:2000 dilution of primary chicken anti-GFP antibody (Aves Labs, GFP-1020, RRID:AB_10000240) for 3 days at 37°C, washed for 1 day (5 ×1 hr), and labeled with a 1:1000 dilution of secondary goat anti-chicken Alexa Fluor 647 antibody (Thermo Fisher Scientific, A-21449, RRID:AB_2535866) for 3 days at 37°C. Due to the fragility of the nasal turbinates housing the majority of zone 1, samples were embedded in 4% agarose (Invitrogen, 16500100) prior to final dehydration (*Figure 1E*). Lastly, the embedded samples were incubated in 66% DCM/33% MeOH for 3 hr, rinsed twice with 100% DCM, and transferred to dibenzyl ether (Sigma-Aldrich, 33630) for final clearing. This experiment was replicated at least five times across multiple cohorts, with each cohort representing all groups.

## Light sheet imaging

Images were collected with a light-sheet microscope (Ultramicroscope II, LaVision BioTec) at ×2.0 magnification using a 640 nm laser and a z-step size of 2.0 µm. All cleared tissue images were acquired with tissue submerged in dibenzyl ether (DBE). Laser intensity was set between 55% and 75% to prevent oversaturated pixels and photobleaching. The working distance of the microscope allowed for complete visualization of both left and right turbinates containing zone 1 OSNs (*Figure 1F and H*). The light-sheet microscope was provided by Cellular Imaging at the Zuckerman Institute (NIH 1S10OD023587-01).

## Imaris 3D cell quantification

All quantification was performed in a double-blind manner. The image stack of GFP+ olfactory sensory neurons in the 647 nm channel was analyzed using Imaris software. The average number of olfactory sensory neurons was measured using the spot detection tool on $350^3$ µm$^3$ cubes of zone 1 tissue, with a requirement of at least 3 cubes per sample for inclusion. The decision to measure an average number of cells within a fixed volume, as opposed to the total number of cells in the turbinates, accounted for potential tissue loss and differences in tissue volumes/shapes across samples. The spot detection was set to a 16.3 µm estimated diameter and was based on Imaris' quality threshold, which compares the intensities at the centers of the candidate spots. The quality threshold varied slightly across samples to adjust for signal quality and axon brightness (to minimize counting spots on axons) but was held consistent within every sample.

### 5-Ethynyl-2'-deoxyuridine (EdU) injections

10 mM EdU (Invitrogen, E10187) was administered to male and female 8–12 week-old M71-GFP$^{+/+}$ and MOR23-GFP$^{+/+}$ mice through a series of daily 0.01 mL/g intraperitoneal injections. Injections were administered 15 min prior to olfactory fear conditioning on each of the 3 days of conditioning, plus 2 additional days around the same time conditioning had been performed (*Figure 3A*). This experiment was replicated at least three times across multiple cohorts, with each cohort representing all groups.

### EdU click chemistry

Mice were transcardially perfused with ice-cold 4% PFA (Electron Microscopy Sciences, 15,710 S) in 1X PBS. MOEs were surgically dissected, incubated in 4% PFA overnight, and cryoprotected in 30% sucrose (Sigma-Aldrich, S0389). MOEs were frozen in OCT (Fisher, 23-730-571) and stored at –20°C until sectioning. Tissue was sliced into 20 μm sections, mounted directly onto Fisher Superfrost Plus glass slides (Fisher, 12-550-15), and stored at –80°C until staining. At the time of staining, slides were acclimated to room temperature, washed with PBST (0.1% Triton X-100 in 1X PBS; 3 × 5 min; Sigma-Aldrich, X100), and incubated with Click-iT Plus EdU reaction cocktail (Alexa Fluor 555; 30 minutes; Invitrogen, C10638). Sections were washed again with PBST (3 × 5 min), with the last wash including 1:10,000 DAPI (Invitrogen, D1306), and then cover-slipped using Vectashield Plus mounting medium (Vector Labs H-1900) and sealed with nail polish.

### Confocal image acquisition

Slides were imaged using a Zeiss Upright LSM 880 Confocal microscope and Zen Black software (Zeiss) or W1-Yokogawa Spinning Disk Confocal (Nikon). All co-labeling images were acquired in z-stacks to ensure accuracy in co-labeling determination.

### Trichamber assay

At least 1 day following olfactory fear conditioning, conditioned mice were assayed in a custom-built acrylic three-chamber box to assess odor avoidance behavior (*Figure 4*). The conditioned and control odors were assigned randomly to either side. Odors were diluted to 1% v/v in mineral oil for all odor preference assays, and flowmeters were set to equal flow rates of approximately 1 l/min. The three-chamber box included vacuum ports both in the center chamber, as well as on either side of the center chamber doorways in the side chambers to ensure that each experimental odor was restricted to its chamber. The mice were habituated to the center chamber for 1 min prior to the start of the test, and then the doors to both chambers were lifted to initiate the assay. Mice were recorded roaming freely throughout the three-chamber box for 10 min. This experiment was replicated at least 5 times across multiple cohorts, with each cohort representing all groups.

### Behavioral analysis

We used ANY-maze software to track the position of the mice throughout the entire trichamber trial. The odor avoidance index was calculated as approach-avoidance index = (time spent on control odor side - time spent on conditioned odor side) / total time spent on either side. Animals visibly sampled each chamber to be included in analysis. We also used ANY-maze to generate heat maps and determine distance traveled, time freezing, and mean speed metrics during the assay. For pose estimation, we trained a SLEAP model (*Pereira et al., 2022*) to track eight keypoints (nose, right ear, left ear, neck, center, right leg, left leg, tailbase) on each mouse across each 10-min trial. With tracking data from 99 (30 F0, 69 F1) 10-min trichamber videos that were all recorded by the same camera and in the same apparatus, we trained a Keypoint-MoSeq model (*Weinreb et al., 2024*) to identify behavioral syllables. The model had six latent dimensions (explaining >90% of the variance) and a kappa value of 1e6. For statistical analyses, a minimum relative frequency cutoff of 0.005 was applied for syllables. Pre-established exclusion criteria required for each mouse to sample each side (control odor and conditioned odor) for at least 3 s for inclusion.

### Statistics

For all F0 experiments, group assignment (naive, unpaired, paired) was randomized. For F1 experiments, group assignment was determined based on the condition of the father used for mating. Power analyses were performed for each experiment using G*Power (3.1) to determine sample sizes.

A significance threshold of p<0.05 was used for all statistical analyses. Outlier tests were performed on all data, leading to the removal of one statistical outlier from the Acetophenone F1 Paired group. This outlier was also excluded from analyses of other metrics from the same behavioral assay. After performing Shapiro-Wilk normality tests, normally distributed data across 2+ groups were performed using a standard one-way ANOVA with Tukey's multiple comparisons post-hoc tests. Normally distributed data between two groups were performed using Student's unpaired t-tests. For data that was not normally distributed, 2+ groups were analyzed with a nonparametric one-way ANOVA (Kruskal-Wallis test) and Dunn's multiple comparisons post-hoc tests. Descriptive statistics used standard error of the mean (S.E.M.) to estimate error. Percent differences between two groups were calculated by comparing the mean of each group. For all analysis except Keypoint-MoSeq, statistical analysis was performed using Prism 9 (GraphPad) software. Statistics for all figures and figure supplements can be found in *Supplementary files 1 and 2*, respectively. For Keypoint-Moseq statistics, a combination of Keypoint-MoSeq code and custom Python code (available on Github at https://github.com/BJMarlinLab/Liff_et_al_2026 copy archived at *Marlin Lab, 2026*) was used for statistical analysis.

## Code availability

Code is available at https://github.com/BJMarlinLab/Liff_et_al_2026 (copy archived at *Marlin Lab, 2026*).

## Acknowledgements

We thank the past and current members of the Marlin Lab for critical discussions and inputs on the manuscript. We thank Dr D Ng and Zuckerman Institute's Genetic 17 Access Tools platform for generating the M71-RFP mouse line. We thank Drs R Axel, S Lomvardas, C Mason, I Abdus-Saboor, I Ahmed, and JJ Marlin for their comments and discussion. We thank AISC, AM., and VLFN for their support. Imaging was performed with support from the Zuckerman Institute's Cellular Imaging platform.

## Additional information

### Funding

| Funder | Grant reference number | Author |
|---|---|---|
| Simons Foundation | Simons Society of Fellows Junior Fellowship 524991 | Bianca J Marlin |
| United Negro College Fund | E.E. Just Fellowship CU20-1071 | Bianca J Marlin |
| Brain and Behavior Research Foundation | NARSAD Young Investigator Grant 30380 | Bianca J Marlin |
| Howard Hughes Medical Institute | Freeman Hrabrowski Scholar | Bianca J Marlin |
| National Institute of Mental Health | T32MH126036 | Clara W Liff |
| National Institutes of Health | 1S10OD023587-01 | Bianca J Marlin |

The funders had no role in study design, data collection and interpretation, or the decision to submit the work for publication.

### Author contributions

Clara W Liff, Conceptualization, Data curation, Formal analysis, Validation, Investigation, Visualization, Methodology, Writing – original draft, Writing – review and editing; Yasmine R Ayman, Eliza CB Jaeger, Data curation, Investigation, Methodology; Avery Cardeiro, Hudson S Lee, Alexis Kim, Angelica Vina-Abarracin, Data curation; Dianne-Lee KD Ferguson, Data curation, Formal analysis; Bianca J Marlin, Conceptualization, Resources, Data curation, Software, Formal analysis, Supervision,

Funding acquisition, Validation, Investigation, Visualization, Methodology, Writing – original draft, Project administration, Writing – review and editing

**Author ORCIDs**
Clara W Liff ⬤ https://orcid.org/0000-0002-2132-3613
Bianca J Marlin ⬤ https://orcid.org/0000-0002-4275-7891

**Ethics**
All animal procedures were performed in strict accordance with the Guide for the Care and Use of Laboratory Animals of the National Institutes of Health. All experiments were approved by the Institutional Animal Care and Use Committee (IACUC) of Columbia University under protocol AC-AABL8552. Mice were housed and handled in accordance with institutional guidelines, and all procedures were conducted to minimize animal suffering.

Reviewer #1 (Public review): https://doi.org/10.7554/eLife.92882.3.sa1
Reviewer #2 (Public review): https://doi.org/10.7554/eLife.92882.3.sa2
Reviewer #3 (Public review): https://doi.org/10.7554/eLife.92882.3.sa3
Author response https://doi.org/10.7554/eLife.92882.3.sa4

---

# Additional files

## Supplementary files
Supplementary file 1. Spreadsheet with all statistics for *Figures 1–6*.

Supplementary file 2. Spreadsheet with all statistics for all figure supplements.

MDAR checklist

## Data availability
Dataset available on Dryad at DOI: https://doi.org/10.5061/dryad.80gb5mm4m. The M71-RFP mouse line is available upon request.

The following dataset was generated:

| Author(s) | Year | Dataset title | Dataset URL | Database and Identifier |
| --- | --- | --- | --- | --- |
| Liff C, Ayman Y, Jaeger E, Cardeiro A, Lee H, Kim A, Vina-Albarracin A, Ferguson D-L, Marlin B | 2026 | Fear conditioning biases olfactory sensory neuron frequencies across generations | https://doi.org/10.5061/dryad.80gb5mm4m | Dryad Digital Repository, 10.5061/dryad.80gb5mm4m |

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
