## [Editor Report · eLife Assessment]

This study provides **solid** evidence that odor fear conditioning biases olfactory sensory neuron receptor choice in mice and that this bias is detectable in the next generation. The authors use rigorous histological and behavioral analyses, including unsupervised behavioral quantification, to support the conclusion that odor-specific sensory representations can be shaped by experience and partially transmitted across generations. While the behavioral effects in offspring are modest and the mechanistic basis of inheritance remains unresolved, the study offers an **important** and carefully executed contribution to understanding experience-dependent sensory plasticity and its intergenerational consequences.

---

## [Referee Report · Reviewer #1 (Public review)]

Summary

The revised manuscript by Liff et al. represents a substantial improvement over the original version. The authors have carefully addressed the key concerns raised in the initial review, most notably by expanding their behavioral analyses and incorporating additional experiments that strengthen the mechanistic links between olfactory sensory neuron (OSN) changes and behavioral outcomes. Their integration of unsupervised Keypoint-MoSeq analysis, extended behavioral metrics (distance travelled, mean speed, freezing time), and the inclusion of behavioral results in the main figures significantly enhance the clarity and impact of the work. The revised discussion also better contextualizes the findings in relation to previous literature, including the discrepancies with Dias & Ressler (2014), and provides more transparency regarding experimental choices.

Overall Evaluation

The revised version has substantially strengthened the manuscript. By addressing the initial concerns with new data, improved analyses, and clearer discussion, the authors provide a much more compelling and rigorous account of how odor-shock conditioning biases OSN fate and influences offspring. Although some questions remain open for future exploration, the present study now makes a clear, well-supported contribution to understanding intergenerational sensory inheritance. I commend the authors for their thoughtful and thorough revisions.

Strengths

Expanded behavioral analysis: The addition of multiple quantitative metrics, inclusion of freezing behavior, and use of Keypoint-MoSeq provide a much richer characterization of behavioral phenotypes in both F0 and F1 generations. These data convincingly demonstrate nuanced odor-specific effects that were not captured in the earlier version.

Improved presentation: Behavioral data, previously relegated to supplementary materials, are now appropriately included in the main figures, supported by supplementary statistical tables. This makes the results more transparent and accessible.

Potential Limitations

Some behavioral effects in the F1 generation remain subtle; the discussion addresses this, but a cautious interpretation of behavioral inheritance would be appropriate.

The MoSeq analysis is a valuable addition, though clarifying what "syllables" represent and how they relate to traditional behavioral measures could aid reader interpretation.

---

## [Referee Report · Reviewer #2 (Public review)]

Summary:

The authors examined inherited changes to the olfactory epithelium produced by odor-shock pairings. The manuscript demonstrates that odor fear conditioning biases olfactory bulb neurogenesis toward more production of the olfactory sensory neurons engaged by the odor-shock paring. Further the manuscript reveals that this bias remains in first generation male and female progeny produced by trained parents. Surprisingly, there was a disconnect between increased morphology of the olfactory epithelium for the conditioned odor and the response to odor presentation. The expectation based on previous literature and the morphological results were that F1 progeny would also show an aversion to the odor stimulus. However, the authors found that F1 progeny were not more sensitive to the odor compared to littermate controls

Strengths:

The manuscript includes conceptual innovation and some technical innovation. The results validate previous findings that were deemed controversial in the field, which is a major strength of the work. Moreover, these studies were conducted using a combination of genetically modified animals and state-of-the-art imaging techniques, highlighting the rigorous nature of the research. Lastly, the authors provide novel mechanistic details regarding the remodeling of the olfactory epithelium, demonstrating that biased neurogenesis, as opposed to changes in survival rates, account for the increase in odorant receptors after training.

Weaknesses:

The main weakness is the disconnect between the morphological changes reported and the lack of change in aversion to the odorant in F1 progeny. The authors also do not address the mechanisms underlying the inheritance of the phenotype, which may lie outside of the scope of the present study.

---

## [Referee Report · Reviewer #3 (Public review)]

Liff et al. have made considerable effort to improve their manuscript. In their revised manuscript, the authors have substantiated their claims of intergenerationally inherited changes in the olfactory system in response to odor-dependent fear conditioning. Several new experiments and analyses now strengthen this study.

I still find that the statement that the study provides "insight into the heritability of acquired phenotypes" is somewhat misleading. In their response to this initially raised point the authors correctly point out that their "results provide basic knowledge that will accelerate our ability to uncover the mechanisms driving heritable changes." That said, current "insights" are not mechanistic in nature.

---

## [Author Response]

The following is the authors’ response to the original reviews.

**Reviewer #1 (Public Review):**
(1) Discrepancies with previous findings need clarification, especially regarding the absence of similar behavioral effects in F1. Lack of discussion on the decision to modify paradigms instead of using the same model. Presentation of behavioral data in supplementary materials, with a recommendation to include behavioral quantification in main figures. Absence of quantification for freezing behavior, a crucial measure in fear conditioning.

We agree, thank you. One of the major revisions we have made to this version of the manuscript is the addition of much more thorough analysis of our F1 behavior. While not captured by the (relatively gross) measure of the approach-avoid index, further analysis has highlighted interesting differences between the F1s of unpaired and paired offspring, and in an odor-specific manner. As these analyses have given rise to many new results and conclusions, we have attempted to adjust the manuscript to reflect the major change that we do, in fact, find effects in F1, if subtle.

Classical odor-shock pairing was used in both Dias & Ressler’s and our study to directly expand upon the findings of increase in cell number. This enabled our discovery of biasing of newborn OSNs. For our behavioral readouts, we chose to focus on the ethological behavior of avoidance. From our extensive behavioral analysis (Figures 5 & 6), we successfully identified several behavioral differences in the F1 offspring that had not previously been described.

**Reviewer #2 (Public Review):**
(1) The main weakness is the disconnect between the morphological changes reported and the lack of change in aversion to the odorant in F1 progeny. The authors also do not address the mechanisms underlying the inheritance of the phenotype, which may lie outside of the scope of the present study.

Thank you for your comments. Our revised manuscript includes both new experiments and new analyses that probe the relationship between a change in cell number and a change in avoidance behavior, and we have revised the manuscript text to address this point directly. In short, we find both in the F0 generation (at extended time points) and in the F1, that an increase in cell number does not always correlate with avoidance behavior. However, we do find nuanced behavioral differences between the offspring of unpaired and paired fathers. Whether the increase in cell number in offspring is necessary to observe the behavioral changes is outside the scope of the current study, but certainly a question we are interested in answering in future work.

**Reviewer #3 (Public Review):**
(1) In the abstract / summary, the authors raise expectations that are not supported by the data. For example, it is claimed that "increases in F0 were due to biased stem cell receptor choice." While an active field of study that has seen remarkable progress in the past decade, olfactory receptor gene choice and its relevant timing in particular is still unresolved. Here, Liff et al., do not pinpoint at what stage during differentiation the "biased choice" is made.

EdU is only taken into stem cells in the S phase, and differences in EdU-labeled M71 or MOR23 OSNs across fear conditioning groups indicates a biasing in subtype identity. We do not make claims regarding the exact stage of OSN maturation at which biasing may occur; rather, we demonstrate that the stem cells that were dividing during EdU administration are more likely to mature into an M71 OSN if a mouse receives paired acetophenone conditioning compared to unpaired or no conditioning (and similarly with MOR23 and lyral). This phenomenon must involve receptor choice, as that is the mechanism by which OSN subtypes form.

(2) Similarly, the concluding statement that the study provides "insight into the heritability of acquired phenotypes" is somewhat misleading. The experiments do not address the mechanisms underlying heritability.

We do not claim to provide direct insight into the mechanisms underlying heritability. Our experiments do provide insight into the heritability of acquired phenotypes, as we corroborate previous studies that this olfactory fear conditioning paradigm induces heritable changes in the nose and in behavior. We also demonstrate odor-specific behavioral differences in the offspring conditioned fathers, suggesting that the mechanisms underlying the specific behavioral phenotypes may be unique to the conditioning odorant, and not one universal mechanism. These results provide basic knowledge that will accelerate our ability to uncover the mechanisms driving heritable changes.

(3) The statement that "the percentage of newborn M71 cells is 4-5 times that of MOR23 may simply reflect differences in the birth rates of the two cell populations" should, if true, result in similar differences in the occurrence of mature OSNs with either receptor identity. According to Fig. 1H & J, however, this is not the case.

We have removed that statement from the manuscript, as subtype-specific differences in proliferation rates are not the focus of this study and we do not wish to make claims about it based on our EdU experiments. We do not compare our iDISCO cell density counts to EdU co-labeling counts nor ratio counts, as differences between M71 and MOR23 quantification in cleared tissue versus EdU uptake may simply reflect the inherent differences between methodologies. Our claims are solely within M71 cohorts and MOR23 cohorts.

(4) An important result is that Liff et al., in contrast to results from other studies, "do not observe the inheritance of odor-evoked aversion to the conditioned odor in the F1 generation." This discrepancy needs to be discussed.

This is discussed in the manuscript, and we report behavioral differences revealed by additional analyses.

(5) The authors speculate that "the increase in neurons responsive to the conditioned odor could enhance the sensitivity to, or the discrimination of, the paired odor in F0 and F1. This would enable the F1 population to learn that odor predicts shock with fewer training cycles or less odorant when trained with the conditioned odor." This is a fascinating idea that, in fact, could have been readily tested by Liff and coworkers. If this hypothesis were found true, this would substantially enhance the impact of the study for the field.

We agree that additional F1 behavioral paradigms are a major next step to understand the functional behavioral differences that may emerge from an increase in specific OSN subtype. Due to the nontrivial amount of time and effort it requires to generate F1 offspring (on the order of many months), and because we do not test individual offspring in multiple behavioral assays (such that they are naïve to their father’s conditioning odor), these experiments are outside the scope of this current study.

**Reviewer #1 (Recommendations For The Authors):**
(1) Considering that the authors are expanding upon the previous findings of Dias and Ressler (2014), it is crucial to clarify the discrepancies in the results between both works in the discussion. While I acknowledge the use of a different experimental design by the authors, if the premise assumes there is a universal mechanism for transgenerational acquired modification it prompts the question: Why don't we observe similar behavioral effects in F1 in the present model? This issue needs extensive discussion in the manuscript to advance the field's understanding of this topic. Additionally, I am also curious about the author's decision to modify the paradigms instead of using exactly the same model to further extend their findings on stem cells, for example. Could you please provide comments on this choice and elaborate on this aspect in the discussion?

We agree, thank you. One of the major revisions we have made to this version of the manuscript is the addition of much more thorough analysis of our F1 behavior. While not captured by the (relatively gross) measure of the approach-avoid index, further analysis has highlighted interesting differences between the F1s of unpaired and paired offspring, and in an odor-specific manner. As these analyses have given rise to many new results and conclusions, we have attempted to adjust the manuscript to reflect the major change that we do, in fact, find effects in F1, if subtle.

Classical odor-shock pairing was used in both Dias & Ressler’s and our study to directly expand upon the findings of increase in cell number. This enabled our discovery of biasing of newborn OSNs. For our behavioral readouts, we chose to focus on the ethological behavior of avoidance. From our extensive behavioral analysis (Figures 5 & 6), we successfully identified several behavioral differences in the F1 offspring that had not previously been described. We have revised the discussion section to elaborate on these decisions.

We incorporated the behavioral data into the main figures and included a freezing metric to Figure 5 (F, J, & N). We did do an analysis of time spent freezing in the control vs. conditioned chamber, but since the F0 paired mice spend so little time in the conditioned odor chamber, they also spend most of their time freezing in the control odor chamber. Thus, we felt it was better to show the overall time spent freezing during the trial.

(2) It is unclear why the authors chose to present all behavioral data to supplementary materials. I strongly recommend not only incorporating the behavioral data into the main figures but also expanding the behavioral quantification. It appears that the author dismissed the potential effects on F1 without a thorough exploration of animals' behaviors. The task contains valuable information that could be further investigated, potentially altering the findings or even the conclusions of the study. Notably, the absence of quantification for freezing behavior is incomprehensive. Freezing is a crucial measure in fear conditioning, and it's surprising that the authors did not mention it throughout the manuscript. I encourage the author to include freezing data in the analysis and other behavioral quantification as follows: (a) freezing during odor presentation and ITI for conditioning days. (b) freezing during odor preference test in all compartments. (c) it is not very clear the design of the Odor preference behavioral testing. Is the odor presented in a discrete manner or the order is constantly presented in the compartment? Could the authors quantify the latency to avoid after the visit in the compartment? (d) in the video it is very clear the animals are doing a lot of risk assessment, this could be also analyzed and included as a fear measure.

Thanks for the suggestion. We incorporated the behavioral data into the main figures and included a freezing metric to Figure 5 (F, J, & N). We did do an analysis of time spent freezing in the control vs. conditioned chamber, but since the F0 paired mice spend so little time in the conditioned odor chamber, they also spend most of their time freezing in the control odor chamber. Thus, we felt it was better to show the overall time spent freezing during the trial. In the methods section we describe that the odor is continuously bubbled into the chamber throughout the trial, but we have clarified this in the main text as well. As for further behavioral metrics like latencies and risk assessment, initial analyses have not shown anything in the F1 data that we wished to report here. Future work from the lab will investigate this further.

(3) In the Dias and Ressler paper, a crucial difference exists between the models that could elucidate the absence of transgenerational effects on F1. In their study, the presence of the unconditioned stimulus (US) is consistent across all generations in the startle task. I am curious whether, in the present study, the authors considered pairing the F1 with a US-paired task in a protocol that does not induce fear conditioning (e.g., lower shock intensity or fewer pairings). Could this potentially lead to an increased response in the parental-paired offspring? Did the author consider this approach? I understand how extensive this experiment can be, therefore I'm not directly requesting, although it would be a fantastic achievement if the results are positive. Please consider discussing this fundamental difference in the manuscript.

To clarify, the F1 generation is presented with the unconditioned stimulus, just never conditioned with it. In these experiments, we were primarily interested in the F1’s naïve reaction to their father’s conditioning odorant, and whether the presentation of that odor in the absence of a stressor would lead to any fear-like behavioral responses.

We have considered the experiments you have suggested and have ongoing projects in the lab further investigating F1 effects and whether their father’s experiences affect their ability to learn in conditioning tasks. Because of the amount of time and effort it requires to generate F1 offspring, and because we do not wish to test individual offspring in multiple assays, we do not present any of these experiments in the current manuscript. Ongoing work is looking into whether 1-day (vs. 3-day) conditioning is sufficient in the offspring of paired mice, and we appreciate the suggestion of subthreshold shock intensity. We will also clarify in the discussion that future work will try to answer these questions.

(4) If the videos were combined it would be better to appreciate the behavioral differences of paired vs unpaired.

Thank you for the suggestion, fixed. Video S1 is now a combination of unpaired and paired example videos.

(5) Figure 3E, is there an outlier in the paired group that is driving the difference? Please run an outlier test on the data if this has not been done. If already done, please express the stats.

We ran an outlier test using the ROUT method (Q=1%) and did not find any outliers to be removed. We also ran the same test on all other data and removed one mouse from the Acetophenone F1 Paired group in Figure 5 (also described in the Methods section).

(6) I understand that using the term "olfactory" twice in the title may seem redundant. However, the authors specifically demonstrate the effects of olfactory fear conditioning. I suggest including "odor-induced" before "fear conditioning" in the title for greater specificity and accuracy. This modification would better reflect the study's focus on olfactory fear conditioning, especially given the authors did not explore fear conditioning broadly (e.g., contextual, and auditory aspects were not examined).

Thank you for your feedback. We found “olfactory” twice as cumbersome. We have changed the title to “Fear conditioning biases olfactory sensory neuron expression across generations”, to more accurately highlight the importance of the olfactory sensory neuron expression, intergenerationally.

(7) The last page of the manuscript has a list of videos (8 videos), but only two were presented.

We have made sure to include all 7 videos (videos 1 and 2 were combined) in this version.

**Reviewer #2 (Recommendations For The Authors):**
(1) The analyses mentioned on lines 210-220 should be presented.

Thank you for the suggestion. We have removed this part of the manuscript as we do not have a large enough n to draw conclusions about cell longevity in this paper. Future studies in the lab will incorporate this analysis.

**Reviewer #3 (Recommendations For The Authors):**
(1) The manuscript contains several supplementary figures and movies that are not referred to in the main text.

All supplementary figures and movies are now referred to in the manuscript text.

(2) In the abstract, the authors state that they "investigated changes in the morphology of the olfactory epithelium." I think that is (technically) not what they did. In fact, the authors do not show any morphometry of the epithelium (e.g., thickness, layers, etc.), but count the density of OSNs that share a specific receptor identity. Along the same lines, the authors state in the abstract that recent work has shown that conditioning is "resulting in increases in olfactory receptor frequencies." However, recent studies did not show increased "receptor frequencies", but changes in cell count. Whether (or not) receptor expression per OSN is also changed remains unknown (would be interesting though).

Yes, agreed. We changed “morphology” to “cellular composition.” We also changed any references to “receptor frequencies” to “olfactory sensory neuron frequencies.”

(3) Reference 20 needs to be updated.

Thank you, updated.

(4) l.52: the distribution of OSNs into (four) zones is a somewhat outdated concept as zonal boundaries are rather blurry. Generally, of course, dorsoventral differences are real.

Yes, we agree and changed the verbiage to “region” as opposed to “zone.” We mainly bring this up because it later becomes relevant that both M71 and MOR23 are expressed in the same (antero-dorsal) region and thus can be quantified with the same methodology.

(5) Fig. 3B & C: the EdU background staining is quite peculiar. Any reason why the epithelium is mostly (with the sustentacular nuclei being a noticeable exception) devoid of background?

We use the ThermoFisher Click-iT Plus EdU kit (Invitrogen, C10638) and it has consistently produced very good signal to noise ratio.

**Responses to Editor’s note**

We thank the editor for their constructive suggestions.

(1) Should you choose to revise your manuscript, please include full statistical reporting including exact p-values wherever possible alongside the summary statistics (test statistic and df) and 95% confidence intervals. These should be reported for all key questions and not only when the p-value is less than 0.05.

Thank you for the suggestion. We created two supplementary tables with statistical reporting: Table S1 for the main figure statistics, and Table S2 for the supplementary figure statistics.